# Optimal deep brain stimulation sites and networks for stimulation of the fornix in Alzheimer's disease

Ana Sofía Ríos[1], Simón Oxenford [1], Clemens Neudorfer[1], Konstantin Butenko[1], Ningfei Li [1], Nanditha Rajamani[1], Alexandre Boutet [2,3,4], Gavin J. B. Elias [2,3], Jurgen Germann [2,3], Aaron Loh[2,3], Wissam Deeb[5,6], Fuyixue Wang [7,8], Kawin Setsompop[7,8,9], Bryan Salvato[10], Leonardo Brito de Almeida[11], Kelly D. Foote [11], Robert Amaral[12], Paul B. Rosenberg[13], David F. Tang-Wai[3,14], David A. Wolk[15], Anna D. Burke[16], Stephen Salloway [17,18], Marwan N. Sabbagh[16], M. Mallar Chakravarty[12,19,20], Gwenn S. Smith [13], Constantine G. Lyketsos[13], Michael S. Okun [11], William S. Anderson [21], Zoltan Mari [21,22], Francisco A. Ponce[16], Andres M. Lozano [2,3] & Andreas Horn [1,23,24] ✉

Deep brain stimulation (DBS) to the fornix is an investigational treatment for patients with mild Alzheimer's Disease. Outcomes from randomized clinical trials have shown that cognitive function improved in some patients but deteriorated in others. This could be explained by variance in electrode placement leading to differential engagement of neural circuits. To investigate this, we performed a post-hoc analysis on a multi-center cohort of 46 patients with DBS to the fornix (NCT00658125, NCT01608061). Using normative structural and functional connectivity data, we found that stimulation of the circuit of Papez and stria terminalis robustly associated with cognitive improvement ($R = 0.53$, $p < 0.001$). On a local level, the optimal stimulation site resided at the direct interface between these structures ($R = 0.48$, $p < 0.001$). Finally, modulating specific distributed brain networks related to memory accounted for optimal outcomes ($R = 0.48$, $p < 0.001$). Findings were robust to multiple cross-validation designs and may define an optimal network target that could refine DBS surgery and programming.

Alzheimer's Disease (AD) is the most common neurodegenerative disease and the fifth leading cause of death in adults older than 65 years with an increasing total healthcare burden currently above $300 billion per year in the US[1], thus, finding effective treatment options for AD has great socioeconomic relevance. The pathophysiology of AD has been associated with amyloid beta (Aβ) protein depositions, phosphorylated tau protein tangles, neuronal and synaptic loss, as well as neurotransmission deficits. Neuronal loss, in particular, results in gross cerebral atrophy with a predilection for structures implicated in memory function, including the Papez circuit[2] and components of the default mode network (DNM)[3,4]. Clinically, these neurodegenerative processes manifest as disturbances in memory, language and executive functions, as well as progressive loss of day to day functional abilities[5]. Given the well described patterns of Aβ[6] and tau depositions[7], these have been the target of therapeutic efforts for over 20 years. However, these efforts have so far been without significant success leading to a range of alternative approaches to modify the disease. One promising approach has been based on observed network alterations throughout the brain[3,4,8], such as decreased connectivity in precuneus, parahippocampal gyrus, thalamus and post central gyrus[9], as well as white

matter disruptions[10], in addition to Aβ and tau patterns[5], which led to conceptualizing AD as a "circuitopathy"[11].

Deep brain stimulation (DBS) has been shown to successfully alleviate symptoms in circuit disorders of the human brain such as Parkinson's Disease[12], Essential Tremor[13], and—more recently—obsessive-compulsive disorder[14] and other neuropsychiatric disorders[15]. In addition to evidence of sensory stimulus producing gamma entrainment and subsequent reduction of amyloid pathology and improvement in spatial and recognition memory in an AD-mice model[16,17], research that investigates neuromodulation in the treatment of AD has accumulated. DBS to the fornix (fx-DBS) has emerged as an investigational treatment targeting associated circuit disruptions with the aim of modulating associative and limbic networks that subserve memory function, and most specifically the Papez' circuit[18]. In addition to evidence of fornix atrophy in mild cognitive impairment (MCI)[19], as a diagnostic or prognostic marker in AD[20–22], and as an essential component of memory formation and consolidation[21], its potential utility as a DBS target was considered after a serendipitous observation of flashback-like episodes during DBS of the hypothalamic region in a patient with morbid obesity[23]. Although the occurrence of such memory events had been reported previously in the context of temporal lobe stimulation[24], the observation of flashback phenomena after hypothalamic region stimulation, in proximity to limbic structures such as the fornix, led to a first series of six AD patients receiving fx-DBS[25]. While alternative DBS target regions including the ventral capsule/ventral striatum[26] and the nucleus basalis of Meynert[27] have been proposed, the fornix has become the most studied region with over 101 patients that underwent this intervention to date[28]. There are now completed phase I[25] and II[29] clinical trials (NCT00658125, NCT01608061), as well as an ongoing international randomized-controlled trial (Advance II, NCT03622905). In addition, recent studies investigated the neural substrates underlying memory flashbacks[30,31] and autonomic response[32] reported in this patient population.

Fx-DBS has been hypothesized to impact circuits by modulating glucose metabolism impairment in temporal and parietal regions, and there is evidence of hippocampal volume increase in mildly affected AD patients after 6 months of stimulation[25]. Nevertheless, the clinical benefits of fx-DBS remain unproven with promising outcomes for some patients, but no benefit for others. Age has emerged as a possible treatment effect modifier in the ADvance trial. Here, among individuals in the early-on arm during phase 1 (but not in phase 2), participants below the age of 65 worsened on the Alzheimer's Disease Assessment Scale−cognitive subscale (ADAS-cog 13) significantly more than older participants[33].

A competing explanation for differences in clinical outcomes across patients could be variance in electrode placement, as demonstrated across multiple disorders treated with DBS[12,14,34–36]. This effect of lead location could be even stronger in investigational DBS targets where the exact target is not yet precisely defined (leading to more variance in placement) and the neural substrates driving clinical outcome remain poorly understood[14,34,36,37].

In the present study we leverage a unique, multi-centric, large dataset (N = 46) of patients treated with fx-DBS (NCT00658125, NCT01608061), to investigate variability in DBS electrode placement applying a state-of-the-art DBS electrode localization method[35] and subsequent DBS fiber filtering[38], sweetspot and network mapping approaches[12] across three levels: (i) effects of focal electric fields on white matter tracts traversing the stimulation volumes, (ii) optimal stimulation sites on a localized voxel level, and (iii) impact of fx-DBS on distributed whole-brain functional networks, identifying (i) the circuit of Papez and stria terminalis (ii) the intersection between fornix and bed nucleus of the stria terminalis and (iii) functional connection to precuneus, prefrontal regions, cingulate, thalamus, basal ganglia and insula as potential drivers of clinical improvement.

## Results

### Patient demographics and clinical results

We performed a post hoc analysis on a series of 46 patients (mean age: $67 \pm 7.9$ years, 23 females) with mild AD (Alzheimer's Disease Assessment Scale 11−cognitive subscale (ADAS-cog 11) of 12−24 points; Clinical Dementia Rating Scale (CDR) of 0.5 or 1.0) who underwent bilateral DBS (electrode type: Medtronic 3387, Medtronic, Minneapolis, MN) targeting the fornix region across seven international centers between 2007 and 2019[25,29], following a standardized stimulation protocol (see supplementary Fig. 1 for patient selection flow, supplementary tables 1 and 2 for inclusion/exclusion criteria, and supplementary table 3 for patients scores). All patients received DBS at a frequency of 130 Hz and pulse width of 90 ms. AD patients had an ADAS-cog 11 score of $18.5 \pm 5.6$ (mean $\pm$ SD) at baseline and $23.6 \pm 10$ one year after stimulation ($-38.6 \pm 48.8$ % change). In each patient, electrode placement was reconstructed using the revised pipeline of Lead-DBS (www.lead-dbs.org[35]). Electrode localization confirmed accurate placement within the ventral diencephalon in all patients (supplementary Fig. 2). However, differences in electrode placement could be observed across patients: 73/92 active contacts featured a radius ≤2 mm to the closest voxel of the fornix, informed by[39]. Similarly, 85/92 active contacts were located ≤2 mm apart from the closest voxel of the Bed nucleus of Stria Terminalis (BNST), informed by[39].

To investigate differential DBS effects on structures more deliberately, electric fields were estimated for the chronic DBS stimulation parameters using a finite element modeling (FEM) approach as implemented in Lead-DBS[35]. Based on the electric field magnitude (E-field), DBS effects were investigated on the white matter (*DBS fiber filtering*[38], Fig. 1a), focal (*DBS Sweetspot Mapping*[40], Fig. 1b), and distributed network (*DBS network mapping*[12], Fig. 1c) level, results were then cross-validated using Leave-one-patient-out and k-fold (3, 5, 7 and 10-fold) designs. For fiber filtering and network mapping, normative connectivity data estimated in healthy subjects was used to define streamlines and regions of interest in this cohort (see supplementary table 4 for underlying data of normative connectomes).

### Tracts associated with optimal DBS response (DBS Fiber Filtering)

As the core analysis of this study, we determined the stimulation of *which fiber tracts* was associated with maximal clinical improvement. This analysis should be seen as the main analysis of the present work since (i) the fornix constitutes a network target aiming to modulate distributed network activity within the circuit of Papez, (ii) the target is a white-matter structure readily identifiable by structural imaging and tractography, and (iii) tractography could be used to define tract-targets in prospective clinical trials, as has been done previously[41]. We applied the DBS fiber filtering approach, introduced in[38] and methodologically generalized for use with E-fields in[42]. While the method has led to robust results that were predictive across DBS cohorts and surgeons in multiple reports and indications[14,38,42], it should still be considered an experimental approach and fiber filtering results hence warrant multiple levels of validations. To do so, patients were first pseudorandomly split into a training (N = 28) and hold-out (N = 18) cohort. We then performed DBS fiber filtering on the training set using an ultra-high resolution normative connectome calculated from a 760 μm resolution whole-brain diffusion scan[43] to identify a set of white matter streamlines connected to the stimulation volume (thresholded E-field following Astrom et al.[44]) of each patient and correlated degrees of overlap with clinical outcome improvements (Fig. 2a, see supplementary table 5 for fiber filtering parameters).

The tracts that accounted for optimal improvement in the training cohort followed the trajectory of the fornix, a parallel bundle ascending from the BNST, which likely corresponds to the stria terminalis, as well as projections connecting to the anterior portion of the thalamus, and an additional anterior orbito-frontal projection.

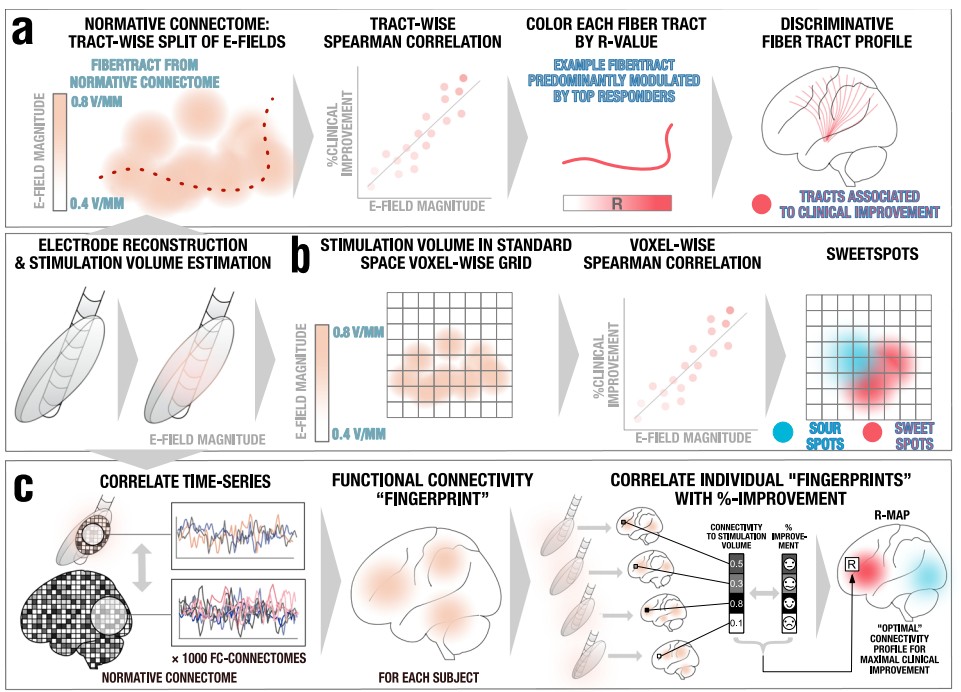

**Fig. 1 | Overview of the three methods applied.** A pre-requisite to run these analyses is to reconstruct the electrode trajectory and localization to then estimate the stimulation volume following the finite element method (FEM). **a** DBS fiber filtering. Stimulation volumes as E-fields were pooled in standard space and overlaid on an ultra-high resolution normative connectome[43]. Peak E-field magnitudes along each tract were aggregated for each stimulation volume and Spearman rank-correlated with clinical outcomes. This led to weights that were assigned to each fibertract. **b** DBS Sweetspot mapping. For each voxel, the E-field magnitudes and clinical outcomes were Spearman rank-correlated, leading to a map with positive and negative associations (sweet and sour spots). **c** DBS network mapping. Seeding BOLD-signal fluctuations from each E-field in a normative functional connectome consisting of rs-fMRI scans from 1000 healthy participants[47] yielded a functional connectivity "fingerprint" map for each patient. Maps were then Spearman rank-correlated with clinical improvement in a voxel-wise manner to create an R-map model of optimal network connectivity.

To validate this set of connections, we first cross-validated the model within the training cohort. To this end, the model was iteratively re-calculated in a leave-one-out design, each time estimating an individual patient's improvement based on the streamlines defined in the remaining cohort (Spearman's R = 0.69 at $p < 10^{-16}$; Fig. 2b). The discrepancy with the actual improvement was quantified using the root means square (RMS = 53.16) and the median absolute error (MAE = 26.07, which is not as susceptible to outliers), shown on supplementary Fig. 3. To further test robustness, we cross-validated the model within the training cohort using k-fold designs that again led to positive and significant correlations between predicted and empirical scores (3-fold: R = 0.52 at p = 0.002; 5-fold: R = 0.58 at $p < 10^{-16}$; 7-fold: R = 0.65 at $p < 10^{-16}$; 10-fold: R = 0.56 at $p < 10^{-16}$, see supplementary Fig. 3 for additional metrics); additionally, 1000 permutations were computed for the training cohort, obtaining an R = 0.69 at p = 0.003 for non-permuted improvement scores (Fig. 2a). This demonstrated high robustness of findings *within* the training cohort. Figure 2b shows two example patients of the *training cohort* with optimal (and correspondingly high E-field overlap with tract model calculated based on all but that one patient) and poor outcome (with minimal overlap), respectively.

Next, we used the fiber model calculated on the entire training cohort (N = 28) to estimate clinical outcomes in patients from the hold-out cohort (N = 18), which had been left as a completely naïve hold-out set. This cross-cohort-prediction revealed a significant relationship (R = 0.45 at p = 0.031, R² = 0.102, RMS = 41.621, MAE = 25.452; Fig. 2c) indicating robustness of the generated model. It should be emphasized that for out-of-sample testing, we calculated the coefficient of determination R² (coefficient of determination) based on the sum of squared errors, and not by squaring the correlation coefficient[45]. Figure 2c again features two example patients—this time from the

hold-out cohort—with either optimal clinical outcome (and correspondingly maximal E-field overlap with the tract model calculated on the complete training cohort) or poor outcome (with minimal overlap), respectively.

As further evaluation, we calculated the predictive tract model based on the training-, hold-out- and combined cohorts, separately. This allowed a direct comparison of results calculated in each cohort by visual inspection, and overlayed the identified bundle with structures of interest from atlases available in MNI space[39,46] (Supplementary Fig. 4). Importantly, we ruled out that this set of connections does not simply represent the average connectivity site from electrodes but indeed a specific subset of connections associated with clinical improvements. This was confirmed by repeating the analysis after permuting improvement values across patients, which isolated different connections in each run (Supplementary Fig. 5), demonstrating the identified and robust set of connections specifically account for improvements following DBS.

As a final validation step, we carried out a leave-one-out cross validation across the entire cohort which yielded an R = 0.66 at $p < 10^{-16}$, RMS = 50.32, MAE = 33.23 between estimated fiber scores and empirical improvements. Further cross-validation k-fold designs led to similar results (3-fold: R = 0.44 at p = 0.002; 5-fold: 0.50 at $p < 10^{16}$; 7-fold: R = 0.48 at p = 0.001; and 10-fold: R = 0.52 at $p < 10^{16}$).

These analyses show robustness and predictive utility of tracts associated with optimal clinical outcomes across cohorts and may constitute a finding of great importance that could influence clinical practice (see discussion), especially with respect to guiding DBS programming after surgery. However, a practical clinical question before surgery is which target coordinate to use during surgical planning. To analyze this question, we carried out a voxel-wise mapping analysis to identify an optimal target sweet spot.

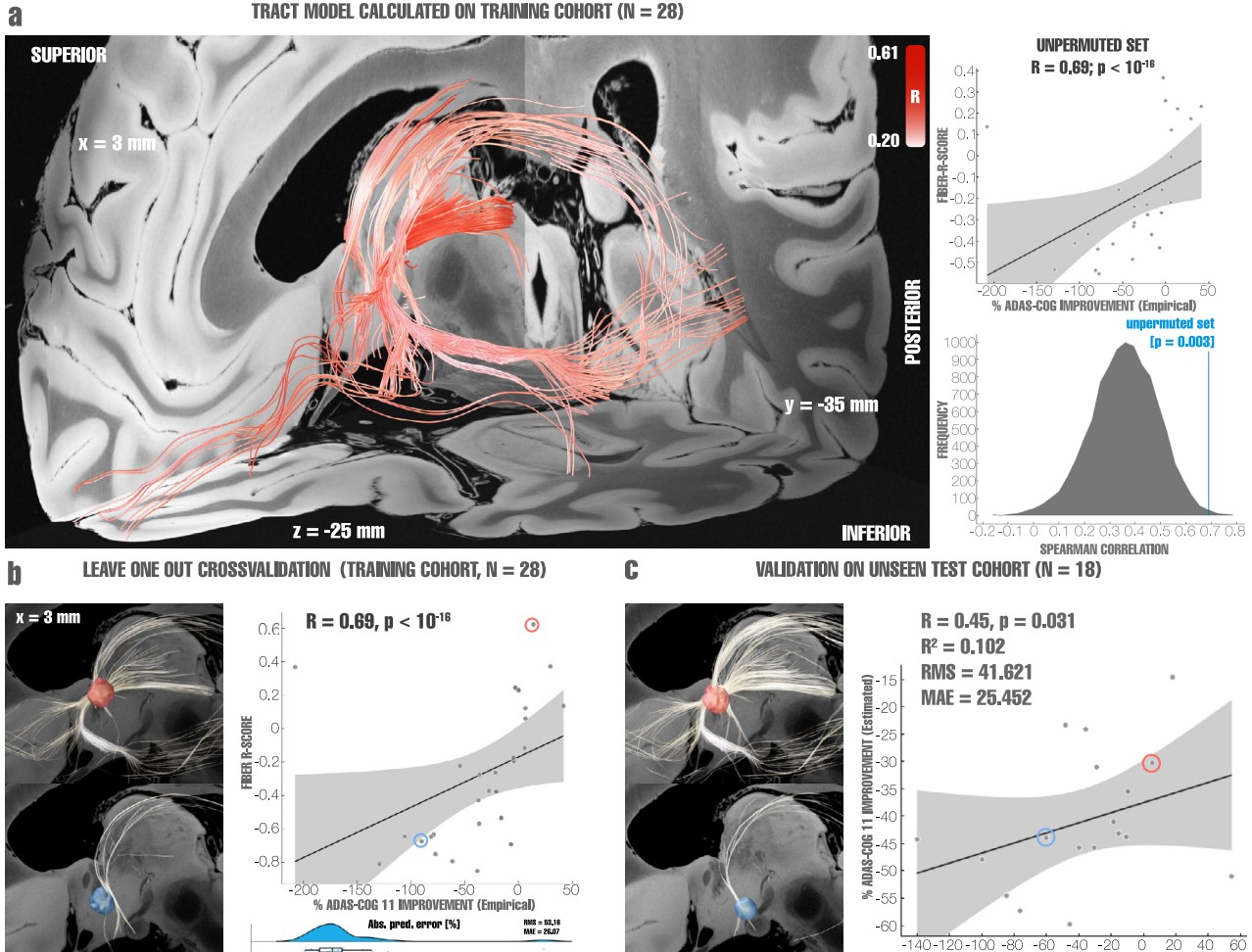

**Fig. 2 | Validation of tract models predictive of clinical improvements as evaluated using ADAS-cog 11. a** Left: Optimal set of tracts to be modulated as calculated from the entire training cohort ($N = 28$ subjects), red intensity codes for R-values ranging from 0.2 to 0.6, with darker colors indicating higher R-values. Right: permutation analysis calculated on the entire training cohort ($R = 0.69$ at $p = 0.003$). **b** Top left: stimulation volume of a patient with top clinical improvement overlapping the tracts associated with optimal clinical improvements (calculated leaving out the subject, $N = 28\text{-}1 = 27$ subjects). Fibers displayed in white correspond to the portion of optimal fibers intersecting with the patient's stimulation volume. Bottom left: Same analysis carried out with a poor-responding example patient. Right: Cross-validation within the training cohort ($N = 28$) using a leave-one-out design (top, $R = 0.69$ at $p < 10^{-16}$), Spearman correlation between the degree of stimulation of positive fibertracts (aggregated R-scores under each E-field) and clinical improvements, and within-fold analysis, reporting root mean square error (RMS) and median absolute error (MAE). The boxplot displays the

interquartile range in the box with the median percentual absolute predicted error as a vertical line, whiskers extend to 1.5 times the interquartile range, outlier points outside of this range are plotted (bottom). The two example patients are marked in the correlation plot with circles. **c** Optimal tracts calculated from the entire training cohort (as shown in panel **a**, $N = 28$) were used to cross-predict outcomes in $N = 18$ patients of the hold-out cohort ($R = 0.45$, $p = 0.031$). Left: two example cases from the hold-out cohort are shown, a top responding patient's stimulation volume with corresponding connected (white) optimal fibers defined by the training cohort; and a poor-responding patient's stimulation volume with corresponding connected (white) fibers. The two example patients are marked in the correlation plot with circles. Right: Spearman correlation between the degree of stimulating positively correlated tracts from the training cohort by the hold-out cohort and clinical improvements of the latter, gray shaded areas represent 95% confidence intervals. Fiber tracts and example stimulation volumes were superimposed on slices of a 100-μm, 7T brain scan in MNI 152 space[83].

## Optimal stimulation site mapping (Sweetspot Analysis)

Sweetspot analysis revealed a consistently symmetric map across the two hemispheres with optimal stimulation sites located at the axial level of the anterior commissure (AC) extending into the descending columns of the fornix bilaterally (Fig. 3). Non-linear flipping of stimulation volumes along the intercommissural plane (which doubles the N of correlations and would be sensible under the assumption of a symmetric DBS effect) led to a similar finding. Peak coordinates and centers of gravity of each cluster are given in supplementary table 6 for both analyses (see also supplementary Fig. 6 for cluster center). The optimal stimulation site was located on the lateral and posterior portions of the columns of the fornix with peak R-values of −0.80 (sourspot) and 0.93 (sweetspot) with unmirrored data and −0.66 (sourspot) and 0.77 (sweetspot) with mirrored data. Note that these

correlation coefficients should not be considered significant due to the mass-univariate (voxel-wise) design. Instead, spatial maps consisting of sweet- and sour-spots were cross-validated across the entire cohort in a leave-one-patient-out design, which led to significant results ($R = 0.33$ at $p = 0.016$, RMS = 50.60, MAE = 27.94). Further cross-validation designs led to similar results (3-fold: $R = 0.27$ at $p = 0.037$; 5-fold: $R = 0.30$ at $p = 0.016$; 7-fold: $R = 0.39$ at $p = 0.005$; 10-fold: $R = 0.33$ at $p = 0.011$).

## Distributed Whole-Brain Networks associated with optimal DBS response (DBS network mapping)

Structural connectivity analyses are limited to identification of monosynaptic connections and probabilistic mapping provides insights on a local level. Hence, in an additional analysis, we investigated modulating

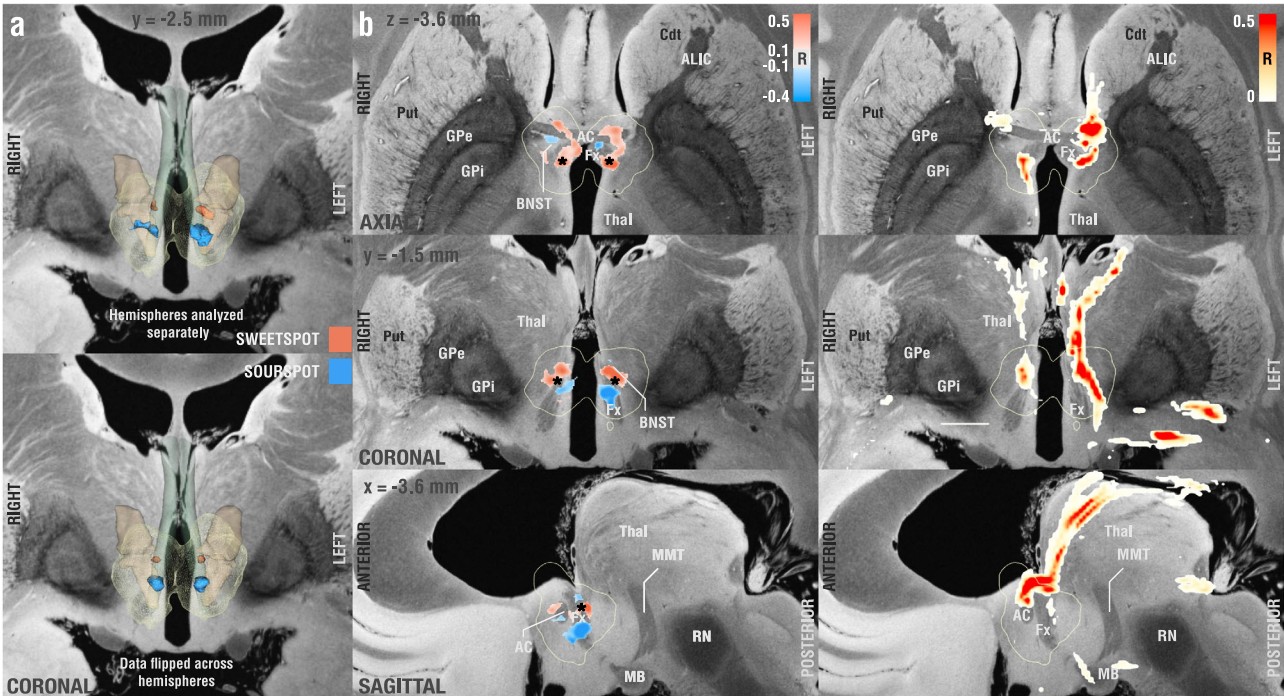

**Fig. 3 | Probabilistic mapping of sweet and sour spots associated with clinical outcome. a** Identified clusters of sweet (red) and sour (blue)-spots in a 3D view, superimposed on slices of a 100-μm, 7T brain scan in MNI 152 space[83]. Since the result was symmetric, on the bottom of the panel, we flipped stimulation volumes across hemispheres to further increase robustness on a voxel-level (effectively doubling the number of electrodes used in each hemisphere). **b** Axial, coronal, and sagittal views of sweet and soursspot peak coordinates (also see supplementary table 6). Projections of cluster center coordinates are marked by a black asterisk

and directly project onto the intersection between fornix and bed nucleus of stria terminalis (BNST, see also supplementary Fig. 6). **c** Axial, coronal, and sagittal sections showing DBS fiber filtering results obtained from the whole cohort at MNI: X = −3.6, Y = −1.5, and Z = −3.6. Put Putamen, Cdt Caudate, ALIC Anterior limb of the internal capsule, AC Anterior commissure, GPe/i external/internal pallidum, Thal thalamus, RN red nucleus, MB mamillary bodies, Fx Fornix. Fornix is shown in blue-green color, informed by the CoBrALab Atlas[46]. Bed nucleus of the stria terminalis shown in light brown color, informed by Neudorfer et al.[39].

which functional whole-brain networks was associated with optimal outcomes. To this end, we applied the DBS network mapping method[12,13] using E-fields as seed regions in a normative connectome calculated from resting-state fMRI scans acquired in 1,000 individuals[47,48] to identify regions correlated and anti-correlated to the stimulation volume area (Fig. 1c). For each patient, this led to a *fingerprint* of functional connectivity seeding from their respective stimulation sites. Voxel-wise values denoted by these connectivity fingerprints were then correlated with clinical improvements following the approach described by Horn et al.[12]. The resulting R-map would show maximal positive values for regions to which connectivity was associated with optimal response, and negative values to regions yielding no clinical benefit (Fig. 4). The map was largely symmetric across hemispheres with R-values ranging from −0.45 to 0.43. Optimal response most strongly correlated with connectivity to precuneus, prefrontal regions, cingulate, thalamus, basal ganglia and insula. To validate these results, we again carried out leave-one-out (R = 0.38 at p = 0.006, RMS = 48.69, MAE = 30.99) and several k-fold cross-validation designs (3-fold: R = 0.32 at p = 0.018; 5-fold: R = 0.14 at p = 0.195; 7-fold: R = 0.44 at p < 10[16]; 10-fold: R = 0.29 at p = 0.026). Moreover, repeating the analysis on the training, hold-out and combined cohorts led to highly similar results by visual inspection (Fig. 4). To allow a certain degree of reverse inference from these network results[49], they were spatially compared to maps associated with a total of 1307 terms present in the Neurosynth database (https://neurosynth.org/)[50]. After excluding purely anatomical/functional terms (such as "prefrontal", "cingulate" or "default"), 7 out of the first 8 cognitive terms related to memory functions or Alzheimer's Disease, namely: "retrieval", "memory", "memory retrieval", "episodic", "task", "demands" and "working memory". The only outlier term not related to memory, "pain", ranked at #5. Functional network

results and their relationship to cognitive terms are summarized in Fig. 4.

Figure 5 summarizes the three levels (fiber filtering, sweetspot mapping, and network mapping, also see supplementary Fig. 7 for in-fold analysis) of analyses across all the patients and the comparable amount of variance explained by each method on a circular (leave-nothing-out) basis, as well as multiple cross-validation designs across the entire cohort. Results (including the same cross-validations) remained highly consistent when repeating all analyses using absolute (instead of relative) improvements on the ADAS-cog 11 scale (Supplementary Fig. 8) and when analyzing the subset of patients enrolled in the ADvance trial (N = 40), in which improvements measured by ADAS-cog 13 were available and applied (Supplementary Fig. 9).

## Effects of age
Prior results had shown differences in clinical improvements related to age groups, where among individuals in the early-on arm during phase 1 (but not in phase 2), participants below the age of 65 worsened on the ADAS-cog 13 significantly more than older participants, while those showed improvement[33]. The robustness of models in the present study to successfully cross-estimate clinical improvements across the entire group regardless of age (and regardless of slicing up the data into leave-one-out, 10-, 7-, 5- and 3-fold cross-validation designs) does not a priori confirm such an effect (i.e., the same model seemed to be predictive in both age groups). An alternate reason for age differences could be (potentially atrophy related) systematic shifts in electrode placements as a function of age. However, as can be seen in supplementary Fig. 10, no apparent difference in electrode placements was observed between the groups, if at all more variability on the z-axis in the young cohort. Furthermore, there was no significant difference in

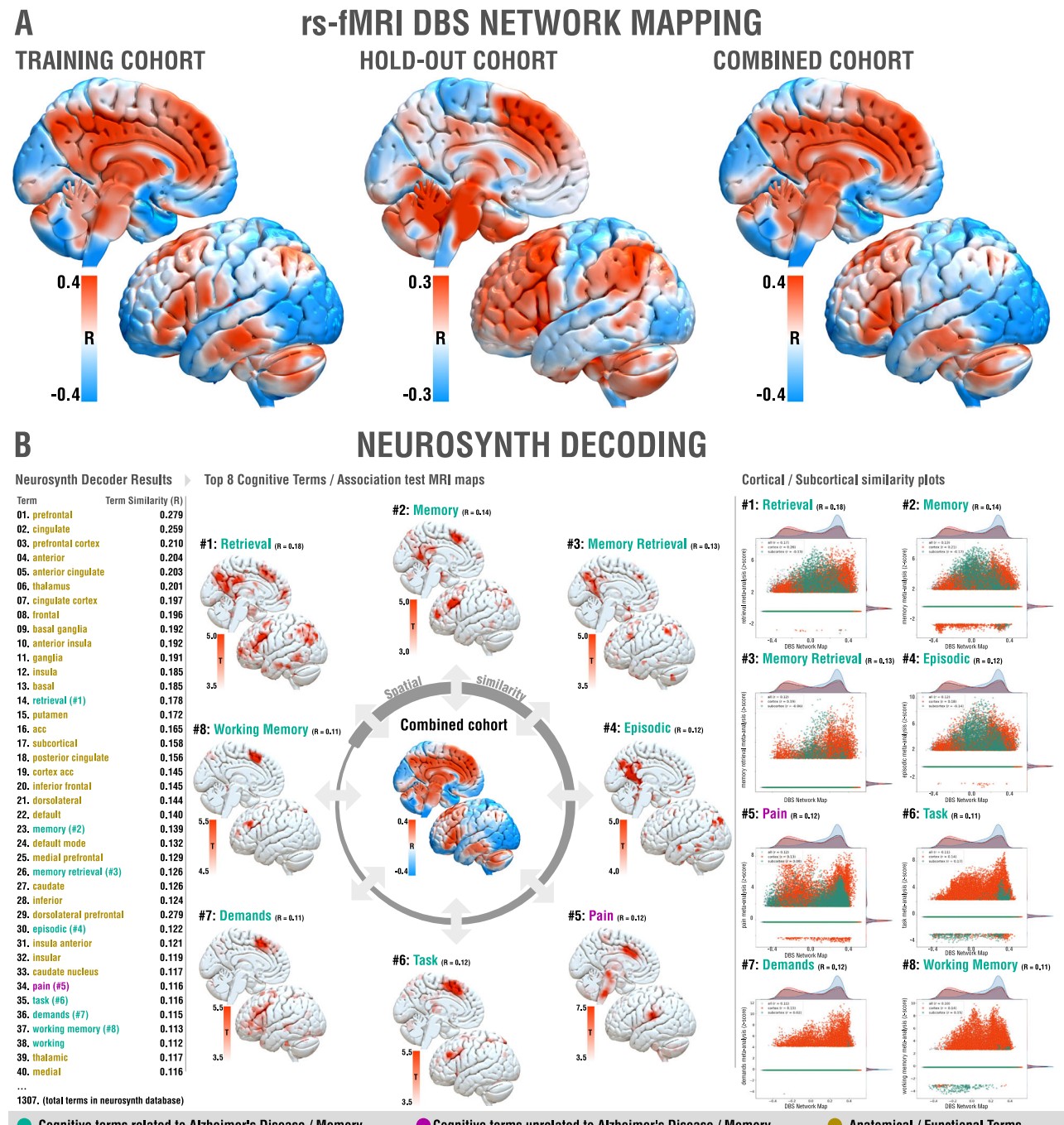

**Fig. 4 | Functional network results. A** Functional networks associated with optimal improvements across training (left), hold-out (middle) and combined (right) cohorts. Brain regions are color-coded by correlations between degree of functional connectivity with DBS electrodes and clinical improvements across the cohorts. Since results were highly symmetric, only the left hemisphere is shown. **B** Optimal network associations to Neurosynth database terms, left: highlighted relevant regions for the most similar networks identified; right: similarity plots between same networks and the optimal network identified by DBS Network Mapping results (x-axis = specific network meta-analysis, z-score, y-axis = DBS Network Map).

fiber scores obtained across the two age groups ($p = 0.790$). This does not suggest a systematic shift between groups (such as stimulation in younger participants systematically modulating optimal fiber connections less strongly than in older participants). Of note, in the present study, both arms of the original study were combined.

**Analysis of Flashback phenomena**

In a sub analysis concerning the original hypothesis that led to fx-DBS in AD, we carried out DBS fiber filtering by contrasting stimulation settings that did or did not induce flashback-like phenomena during

the surgical procedure[30,31]. On a localized level, this effect had been studied before[30,31], but not on a tract level. The sub-cohort in which this information was available included 39 patients in which different DBS parameters were probed, leading to a total of 2054 stimulation volumes, of which 66 resulted in experiential flash-back episodes. In contrast to clinical improvements, flashback-like phenomena were significantly associated with modulation of the posterior limb of the anterior commissure (Fig. 6), which interconnects the middle and inferior gyri of the bilateral temporal lobes[51]. Critically, electrical stimulation of these cortical regions has been associated with

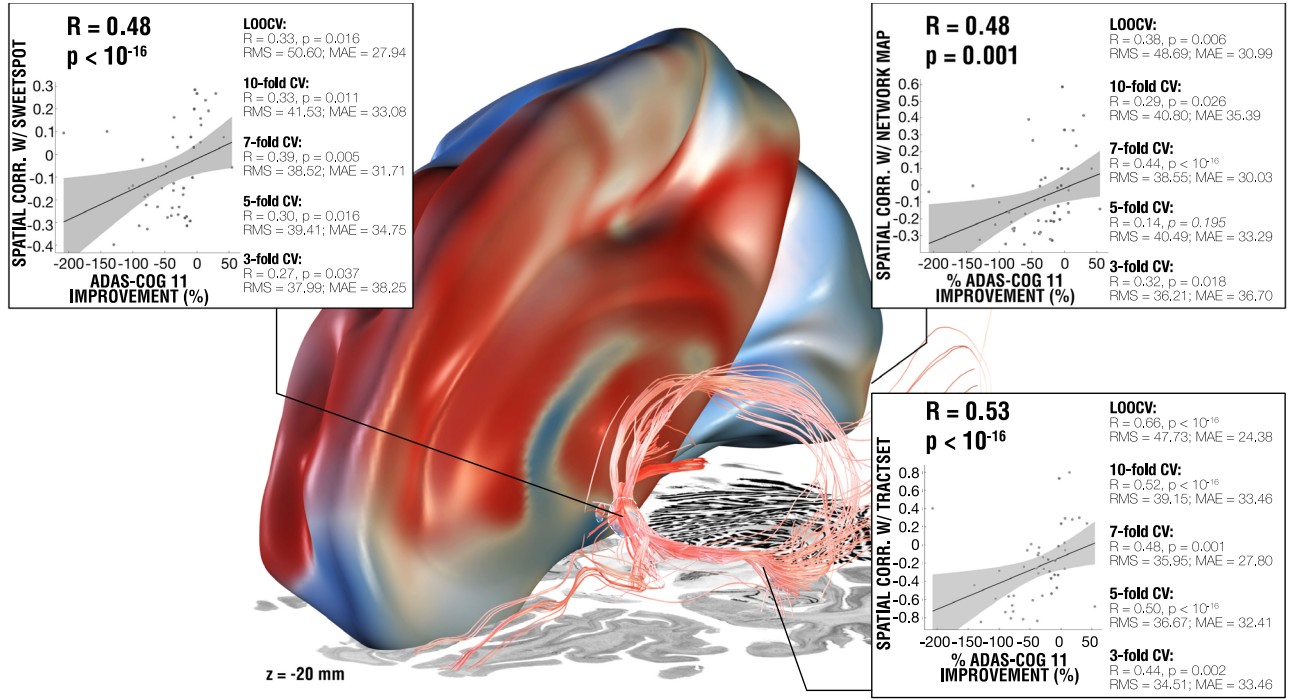

**Fig. 5 | Results summary including the models from DBS fiber filtering, sweetspot mapping and network mapping.** The three levels of analysis were able to explain a similar amount of variance of clinical outcomes when analyzed in a circular nature (see scatterplots; ~16–19%) and led to significant cross-predictions of clinical outcomes across leave-one-patient-out and multiple k-fold designs, plots show fitting of a linear model that represents the degree to which stimulating voxels (left), functional regions (top-right) and tracts (bottom-right) explain variance in clinical outcomes across the whole cohort ($N = 46$) using Spearman correlation, gray shaded areas represent 95% confidence intervals. Three level analysis results were superimposed on slices of a brain cytoarchitecture atlas in MNI 152 space[84]. See supplementary Fig. 7 for additional metrics on each validation approach. RMS Root mean square error, MAE Median absolute error.

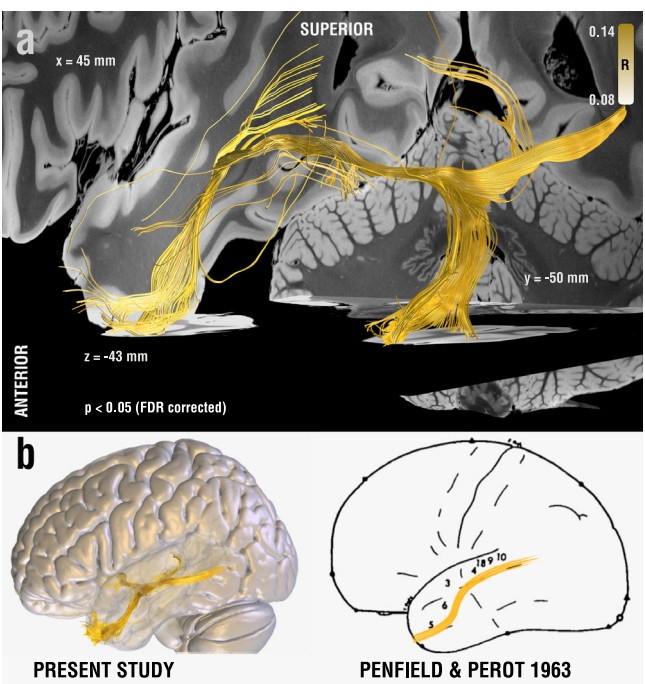

**Fig. 6 | White matter bundle associated with occurrence of flashback-like phenomena. a** Fiber tracts correlated to the presence of flashback-like events, connected fibers were corrected for multiple comparisons using the False Discovery Rate (FDR) at a 5% α-level. **b** Brain surface (lateral view) overlaid with results from **a** (left), in comparison to Penfield's original work on mapping the presence of electrical stimulation-induced "experiential phenomena" in 40 patients suffering from temporal lobe seizures in a total of 1288 reviewed surgical cases covering a large fraction of the cortical mantle (right). Adapted with permission from[24].

flashback-like phenomena in multiple historical and contemporary reports[24,52].

## Discussion

A three-level post hoc analysis, at the local, structural, and functional connectivity level, was carried out in a cohort of 46 mildly affected AD patients treated with fx-DBS across seven international centers. The results obtained from these analyses provide insights into (i) the fiber tracts associated with optimal outcomes, (ii) optimal stimulation coordinates (sweetspot maps), and (iii) functional whole-brain networks associated with optimal outcomes.

There were many factors that could have led to variability in DBS electrode placements within the fornix region across cases and centers. These factors included decreased fornix volume in AD or its preclinical stage MCI[19], the complexity of reaching this target using electrodes (transventricular approach)[53] and the possible variations in placement due to surgeon to surgeon variability. Consequently, electrode localizations varied within the diencephalic region. Considering these variations in DBS lead placement, we sought to examine which white matter pathways were modulated in top-responding but not poor-responding patients. We addressed this question using the DBS fiber-filtering method on tracts defined by an ultra-high-resolution connectome, which was acquired at an isotropic 760 μm resolution, and contains proper definition of fine bundles (such as the stria terminalis) frequently obscured in single-patient scans. Given the historical development of fx-DBS, we hypothesized that fiber tracts associated with optimal response would include memory-relevant connections, specifically the structures of the Papez' circuit, whose role is crucial in episodic memory[54] and for which changes have been described as early as in prodromal AD or MCI. This hypothesis was supported by our analyses. Indeed, both fornix and stria terminalis were strongly associated with optimal clinical response. Given the

strong implications for clinical practice that our results might have, we cross-validated results on multiple levels, which could demonstrate remarkable consistency of findings throughout subsets of the entire cohort. Based on this white matter model, we were able to estimate a significant amount of variance in clinical outcomes both within the $N = 28$ training-cohort (leave-one-out and several k-fold designs), and when estimating clinical outcome of patients in the hold-out-cohort ($N = 18$) based on the model calculated exclusively from the training cohort. Finally, cross-validations across the $N = 46$ combined cohort (leave-one-out and several k-fold designs) again showed consistency of findings. Predictive fibers calculated on training and hold-out cohorts alone were remarkably similar, each suggesting a strong involvement of fornix, anterior nuclei of the thalamus and stria terminalis. Interestingly, our analysis yielded a distinct set of streamlines when investigating the presence of flashback phenomena reported during postoperative stimulation programming[30,31]. Here, the posterior limb of the anterior commissure emerged as a substrate of modulation. This result supports the main findings from Germann et al.[31], previously associating stimulation of the anterior commissure with the occurrence of flashbacks, using a different methodological approach.

A seminal historical article by Wilder Penfield and colleagues[24] associated electrical stimulation of specific sites of the temporal cortex with the occurrence of flashbacks, and this has been recently confirmed by other studies[52]. While the thinner anterior limb of the anterior commissure connects bilateral anterior olfactory nuclei and the primary olfactory cortices, its thicker posterior limb connects the bilateral temporal regions associated with the flashback phenomena reported by cortical stimulation studies[24,52]. Hence, a direct modulation of temporal cortices and/or their network with other structures connected to the anterior commissure might provide a potential reason for the occurrence of flashback phenomena following DBS to the fornix region (i.e., the effect is mediated by modulating the AC, not the fornix itself).

These tract-level results enhance our understanding of fx-DBS. However, in surgical decision making, defining a focal optimal stimulation coordinate or region could provide additional practical relevance. Hence, we performed a focal analysis to identify a specific sweet spot target associated with clinical improvement. Original surgical coordinates of the active contacts as described for the surgical approach by Ponce et al.[53] corresponded to an MNI space coordinate[55] of $x = \pm 7.02 \pm 0.68$ mm, $y = 0.82 \pm 1.00$ mm and $z = -6.43 \pm 0.75$ mm. In our analysis, we found that cluster centers of positive correlated voxel values instead corresponded to x = $\pm 4.8$, $y = -0.9$ mm and z = $-3.6$ mm, with a Euclidean distance of 3.87 mm to the original target site. Expressed in functional (AC/PC) coordinates, our target would correspond to a coordinate $5.56 \pm 0.88$ mm lateral to, $-2.87 \pm 0.91$ mm anterior to and $0.65 \pm 1.19$ mm below the anterior commissure[55]. Anatomically speaking, the optimal site corresponded to the border between BNST and fornix at a superior (AC level) and posterior portion of the fornix (Fig. 3 and supplementary Fig. 3). Our data suggest that coordinates located more superiorly, and slightly more medial to the current target, may result in better clinical outcomes, a possible explanation might be that the E-field generated when neuromodulating inferior regions of the fornix could be reaching other structures capable of causing side effects, for instance, autonomic responses after hypothalamic nuclei stimulation[32].

We next applied DBS network mapping using a normative functional connectome to study the relationship between modulating distributed whole-brain networks and clinical improvements. In patients with an optimal cognitive response, DBS stimulation sites fell onto a network comprised of regions of the default mode network (especially the precuneus), previously associated with AD pathology[3,4]. Furthermore, the network included premotor cortical sites involved in memory, working memory and retrieval. A common mistake in the fMRI literature is to infer the actual cognitive function from activation

(or connection) sites[56,57]. This process, termed reverse-inference, is controversial since activity in most brain regions is non-specific across cognitive domains. For example, Broca's area is involved in language processing, but also in other forms of hierarchical processing such as mathematics or music[58]. Hence, (reversely) inferring from an activation in Broca's area that language is involved would be considered a sub-optimal approach[56,57]. To account for this, the creators of the *neurosynth* platform devised a decoding tool facilitating a systematic comparison of network maps with a large amount of meta-analytic maps ($N = 1307$ at the time of writing) associated with specific cognitive terms. Each of these maps represents automatic meta-analyses that often rely on a high number of studies—for instance, the map associated with the term memory is currently based on 2744 studies. The decoding tool compares spatial similarity of a given network with all maps in the database, and sorts resulting spatial agreement with term names in descending order. In our case, the functional network most associated with optimal outcome best resembled the maps built from cognitive terms such as "retrieval" or "memory", hence demonstrating a certain specificity of the identified optimal stimulation network to memory retrieval.

All three levels of analysis (local, tract and network) were highly robust towards multiple cross-validation designs (summarized in Fig. 5, in-fold analysis summarized in supplementary Fig. 7). The findings of this study provide a framework for the neural substrates implicated in successful fx-DBS and offer the potential to refine and guide both surgical targeting and stimulation optimization in Alzheimer's disease in future trials.

## Limitations

Multiple limitations apply to this work, including the retrospective nature of the study, due to which a detailed focus on specific clinical effects was not possible. For this reason, we considered clinical outcomes as measured by change of the ADAS-cog 11 score but repeated main results for ADAS-cog 13 scores in the subset of patients in which the score was available (Supplementary Fig. 9). The retrospective nature of our study also prevented us from analyzing different effects of stimulation frequencies, pulse widths, or stimulation patterns, which would enfold different signals onto the network over time. Instead, the imaging nature of our study analyzes results in static fashion (both on a stimulation volume and network level). Future research is needed to investigate effects of variations in stimulation parameters, such as the ongoing trial to optimize electrical stimulation parameters of fornix-DBS for AD (NCT04856072). Alternatively, neuromodulation delivered through distinct approaches, namely, the ongoing trial on gamma entrainment via sensory stimulus at a 40 Hz frequency (NCT04055376) could extend our knowledge on the effect of diverse parameters in brain stimulation for AD.

An inherent limitation of studies as the present one is imaging resolution and resulting inaccuracies of DBS mapping in standardized stereotactic space, which implies co-registration inaccuracies[59]. This inaccuracy could be even more pronounced in AD patients, who characteristically feature structural changes in both white and grey matter, particularly in early onset AD[10,60,61]. To address these issues, a modern DBS imaging pipeline[35] with advanced concepts such as brain shift correction[62], multispectral normalization[63], and phantom validated electrode localizations[64] was applied. Each processing step was meticulously monitored and corrected, if necessary. In addition, we applied a recently introduced manual refinement of normalization warp fields[65], which was crucial to yield accurate registrations due to large variabilities in patient anatomy. A demonstration of this labor-intensive manual refinement process is visualized in supplementary video 1, which shows that upon manual refinements, a good registration accuracy between patient and template fornices was achieved. In this regard, we were not able to find apparent differences in (i) electrode placement or (ii) fiber-score activations between patients

younger than versus older than 65, which suggests other factors might have influenced the clinical outcome in the younger group (Supplementary Fig. 10). As previously reported, possible explanations for the decline in early onset subjects include a more aggressive presentation of the condition, greater brain atrophy and comparably more reduced glucose metabolism in this subgroup of patients[29].

Another limitation was the combination of randomized and open label outcome data. Due to the exploratory feature of this analysis and aiming at robustness of results, our cohort included patients from different studies, namely a phase I study and the randomized phase II ADvance trial. The inclusion of these two cohorts made it possible to have a large enough sample size to leave a naïve subset of patients to cross-predict our fibertract model. Nevertheless, this sample size is considered small for machine learning approaches, thus, rigorous evaluation was performed to the results presented in this work, including cross-validation at several levels.

Moreover, we must emphasize that conclusions about connectivity profiles associated with optimal outcomes were based on normative connectivity data acquired in healthy participants. While this concept has led to meaningful and robust models in other cohorts[12-14,38,66], conclusions about networks prevalent in the individual DBS patients may not be drawn. However, models describing optimal connectivity based on normative vs. disease-matched vs. patient-specific data were comparable in other diseases, such as Parkinson's Disease and OCD[38,67]. In the present study, electrodes were placed within the diencephalic region, a region featuring complex neuroanatomical relationships and a multitude of intersecting and delicate fiber bundles. Hence, it was a crucial pre-requisite of the study to use a tractogram that exhibits small fiber bundles in accurate anatomical detail. We used a normative whole-brain connectome calculated from an unprecedentedly high-resolution in-vivo dMRI dataset that was acquired across a total scan time of 18 h at 760 μm isotropic resolution on specialized MR hardware[43], as for network mapping, a connectome obtained from rs-fMRI data from 1000 healthy subjects was used to inform regions co-activated with the stimulation volumes of each patient, allowing an identification of circuits that could be involved in clinical changes when modulating the fornix.

Based on three levels of analysis, our results point towards a potential optimal stimulation target for Alzheimer's Disease treatment with fx-DBS. At a local level, our findings highlight a circumscribed region at the intersection of fornix and bed nucleus of the stria terminalis. We further showed that optimal tract connections to this region contained within the circuit of Papez were important, while flashback phenomena were associated with modulating the posterior limb of the anterior commissure. Finally, our results suggest that modulating specific whole-brain networks is crucial for DBS induced positive effects on cognition. Though our data identified a specific site for stimulation, we would like to emphasize that the use of indirect coordinate systems for DBS targeting is not suitable for DBS to the fornix region in patients with atrophy in the same region. Direct imaging and fiber-tracking results will be important to determine accurate targeting in this region.

## Methods
### Patient cohort and Imaging
We conducted a secondary post-hoc analysis of data from a sample of 46 patients (mean age: 67 ± 7.9 years, 23 females), with a clinical diagnosis of mild probable AD that underwent bilateral DBS to the fornix at seven international centers included in the ADvance trial (NCT01608061)[33] and the Toronto-based pilot trial (NCT00658125)[25], all procedures were carried out according to the declaration of Helsinki from 1975, all participants signed an informed consent in person with the participation of a surrogate consenter. While the ADvance trial included 42 patients, imaging data was only available for 40 patients[31,32] (also see supplementary Fig. 1). Patients were diagnosed by

standardized criteria after expert examination rated with 0.5 or 1 on the Clinical Dementia Rating scale (CDR) and scored 12-24 on the Alzheimer's Disease Assessment Scale 11−cognitive subscale (ADAS-cog)[68], further inclusion and exclusion criteria for the trials can be found in supplementary tables 1 and 2, patients received monopolar stimulation at a frequency of 130 Hertz with a 90 microsecond pulse width for 12 months without adjustment. Patients included in the ADvance trial were evaluated using the ADAS-cog 13 scale but for remaining patients, improvements along ADAS-cog 11 was available. Hence, for consistency across the entire cohort, two tasks were excluded from this scale (number cancellation and delayed free recall tasks)[69], and only tasks included in ADAS-cog 11[70] were included for analysis. We repeated main analyses using ADAS-cog 13 in the subset of patients in which the score was available. Patients underwent surgery targeting the descending columns of the fornix using quadripolar electrodes (Medtronic 3387, Medtronic, Minneapolis, MN). $T_1$- and $T_2$-weighted volumetric pre- and postoperative scans obtained at 1.5T across seven sites were used. Intra- and post-operative test stimulation observations and individual stimulation parameters including electrode contact, stimulation amplitude, frequency, and pulse width were included. The additional post hoc data analysis carried out in the present study was approved by the ethics board of Charité−Universitätsmedizin Berlin (master vote EA2/186/18). The clinical outcome of all patients was evaluated using the ADAS-cog 11[70] measured before and one year after the onset of stimulation. Exclusively for means of visualization, participants were classified according to their ADAS-cog 11 outcome as poor responders (decrease of 21% or more), middle responders (0-to--20.99% decrease), top responders (increase in ADAS-cog 11 score percentual change), as shown in supplementary table 3. This classification was not used for statistical analyses, which were carried out on the continuous outcome variable (percentual outcome on ADAS-cog 11 one year after stimulation onset) and the discrete variable TEMPau score[71], used to estimate flashback episode intensities.

### DBS electrode localization and stimulation volume (E-field) estimation
Image pre-processing, electrode localization and estimation of stimulation volume were carried out using default parameters in Lead-DBS[35,62] (www.lead-dbs.org). Briefly, post-operative MRI scans were linearly co-registered to preoperative T1 images using Advanced Normalization Tools[63] (ANTs; http://stnava.github.io/ANTs/). Subcortical refinement was applied to correct for brain shift. Co-registered images were then normalized into ICBM 2009b Nonlinear Asymmetric ("MNI") template space using the SyN approach implemented in ANTs, with an additional subcortical refinement stage to attain a most precise subcortical alignment between patient and template space ("Effective: Low Variance + subcortical refinement" preset). While this method has been shown to yield the best performance for subcortical image registrations[72], the substantial atrophy in this particular population resulted in suboptimal automatic registration results. For the present study, this was crucial, since in the field of DBS, electrode displacements of a few millimeters will lead to substantially different effects[35,37]. To account for this, we applied a method, termed WarpDrive[65], to manually refine registrations into template space (see supplementary video 1). Briefly, WarpDrive provides a graphical interface allowing precise alignment of source and target landmarks by directly visualizing the normalized images, together with the template and atlases in MNI space (the software is openly available here: https://github.com/netstim/SlicerNetstim). WarpDrive allows the user to manually correct misalignments from the standard normalization and recomputes a refined deformation field in real time. DBS electrodes were pre-localized using the TRAC/CORE algorithm[62] and manually refined if necessary. Stimulation volumes were estimated using the finite element method (FEM) within

the adapted FieldTrip/SimBio pipeline[73] (https://www.mrt.uni-jena.de/simbio/; http://fieldtriptoolbox.org/) implemented in Lead-DBS[35]. In brief, a volume conductor model was constructed based on a four-compartment mesh that included gray and white matter, electrode contacts and insulating parts. Gray matter structures were based on an atlas of the human hypothalamic region[39]. The electric field (E-field) distribution was then estimated by solving Laplace's equation for the static approximation of Maxwell's equations on a discretized domain represented by the tetrahedral four-compartment mesh. For the purpose of this article, we occasionally use *E-field* as shorthand for the voxelized magnitude of the electric field vector. The stimulation volumes were defined as *thresholded* versions of the E-field magnitude following the approach in[44].

### Modeling considerations

Estimated after Pakkenberg and colleagues[74], each cubic millimeter of cortex is filled with ~170,000 neurons; for axonal numbers, each fiber bundle in a standard neuroimaging analysis represents $10^3$–$10^5$ tightly packed axons[75]. Many DBS studies aimed at modeling discretized and realistic axonal cable models, in the past[59,76,77]. However, given these sheer numbers of axons involved, here, we chose to assume prob-abilistic axonal populations in each brain voxel to be represented by each fiber tract, instead of modeling representative single axons. While single axons fire in an all-or-nothing fashion, activation/modulation profiles of axonal populations within a voxel may be represented in probabilistic fashion, which would be dependent on the applied voltage[78–80]. In other words, on a population level, the "degree" of activation will be stronger under higher voltages applied, i.e., closer to the electrodes. Crucially, there is a large amount of uncertainty about this exact relationship between voltage and population-level axonal firing that needs patient-specific calibration even when applying more realistic biophysical models[76]. To account for this uncertainty, we applied Spearman's rank correlations to our fiber filtering and optimal stimulation site mapping models (Fig. 1). We propose that this simple approach could be advantageous, since it would show maximal correlations for any type of monotonically increasing dose-effect function. In other words, the concept could be robust toward the exact relationship (be it e.g., linear, cubic, or logistic) between amplitude and axonal modulation.

For each of the models, the stimulation volume of each patient was considered the core of the analysis; for fiber filtering, streamlines from a normative structural connectome that traversed the volumes were considered for further steps; for sweetspot analysis, areas of interest were determined based on voxels occupied by stimulation volumes of the patients; finally, for network mapping, functionally connected areas to the stimulation volume of each patient were obtained from a functional normative connectome. Details for each method are specified in the following sections.

### DBS fiber filtering

Model definition (Fig. 1A): Whole brain structural connectivity profiles seeding from bilateral E-fields were calculated using a state-of-the-art multi-shell diffusion-weighted imaging dataset acquired across 18 scanning hours of a single individual at 760 μm isotropic resolution[43] using the generalized q-sampling approach (default parameters) and whole-brain tracking (default parameters) as implemented in DSI studio[81]. The patients were distributed into two cohorts: Training ($N = 28$) and Hold-out ($N = 18$). For each subject of the training cohort, fibers traversing each voxel of the E-field were selected from the 5 million tracts in the normative connectome and projected to a voxelized volume in MNI space. Each of these fibers were weighted according to the E-field magnitude at each voxel, considering only fibers that traversed > 20% of stimulation volumes with an E-field magnitude > 0.36 V/mm. Each fiber was then appointed an R-value dependent on the Spearman correlation between its weighting and the

respective clinical outcome scores across the group, i.e., a high R-value indicates that the modulation of the tract is associated with clinical improvement. Given the mass-univariate nature of this approach (and subsequent alpha-error accumulations), the resulting correlation coefficients were not considered significant, but were rather used to discriminate and visualize a specific set of bundles that was later validated by estimating clinical outcome in out-of-sample data (Fig. 1, supplementary Table 5).

Estimating outcomes using the model: Assuming a patient would most likely show superior clinical benefit if their E-field modulated more fibers with high positive R-values and less fibers with negative scores, we measured the spatial Spearman's rank correlation profile of the (hold-out) E-field superimposed to the tract model. To illustrate by an example: If an E-field peaked at sites coinciding with tracts with high positive R-values and showed low amplitudes at sites filled by tracts with low R-values, this would lead to a high Fiber-Score for that particular E-field.

Cross-validation and testing: We first estimated our model by defining a "Training cohort" including 60% of the participants in a pseudorandomized fashion, and filtering fibers with positive R-values across this group; the remaining 40% of the participants ("Hold-out cohort") were left to validate the predictive utility of the model. The Training cohort was used to estimate an optimal connectivity model. In an initial training stage (using only data from the training cohort), model parameters were still manually tuned using the graphical user interface created for Fiber Filtering within Lead-DBS. Aims were to obtain a set of fibers that was (i) robust for cross-validations and (ii) variable when permuting improvement values across patients. The latter point was crucial, since specific parameter settings exist that would result in a set of tracts that were simply connected to the average group of electrodes. In such settings, permuting improvement values across the cohort would not largely alter results. After several iterations, settings were obtained (supplementary table 5) that fulfilled both criteria and showed robust cross-validation results (leave-one-out and multiple k-fold [k = 3,5,7,10] designs). Then, model parameters were kept fixed and the model was used to cross-predict outcomes of patients in the hold-out cohort.

### Optimal stimulation sites (Sweetspot analysis)

Model definition (Fig. 1B): Using the E-fields calculated for each patient, an approach to define optimal stimulation sites was applied[12]. An E-field represents the first derivative of the estimated voltage distribution applied to voxels in space and its magnitude is hence stronger in proximity of active electrode contacts with a rapid decay over distance. Since not all voxels were covered by the same number of E-fields, the area of interest was restricted to voxels that were at least covered by 20% of E-fields with a magnitude above 200 V/m, which is a common approximate assumed to activate axons in the field of DBS[42]. For each voxel covered by the group of E-fields across the cohort in MNI space, E-field magnitudes across patients were Spearman rank correlated with clinical outcomes. The resulting sweetspot maps would peak at voxels in which stronger E-fields were associated with better treatment responses. The map would have negative values for voxels with the opposite relationship.

Estimating outcomes using the model: Multiplying each voxel of a single E-field with the resulting sweetspot map and calculating the average across voxels led to estimates of how a specific E-field would perform (i.e., estimates of clinical outcomes following DBS). If the E-field peaked at similar locations as the sweetspot map, a high estimate would result. If it peaked at a valley of the map, low or even negative estimates would result. The values of these maps were analyzed using Multi-image Analysis GUI software[82] (http://ric.uthscsa.edu/mango/) to estimate the peak and center location of clusters in both positive (sweetspot) and negative (sourspot) correlated voxels, this analysis was repeated in E-fields mirrored to opposite hemispheres

to obtain a more robust observation of peak voxels. Again, cross-validation of the model was carried out by means of a leave-one-out and several k-fold [$k$ = 3,5,7,10] designs.

### DBS network mapping

Model definition (Fig. 1C): In a third approach, we calculated whole-brain functional connectivity estimates seeding from E-fields using a normative connectome that was calculated from rs-fMRI scans acquired in 1000 healthy participants, providing a map of coupling brain regions based on their blood-oxygen-level-dependent (BOLD) signal[47,48], following the approach developed by Horn et al.[12]. This method, termed DBS network mapping, allows to investigate functional connectivity profiles of a specific pair of DBS electrodes. We refer to the maps resulting from an estimation of correlated "active" brain regions seeded from each stimulation volume using normative data as *connectivity fingerprints*[66]. Similar to the sweetspot and fiber filtering models, (voxel-wise) correlations between Fisher-z-scored connectivity strengths and clinical improvements were calculated, which yielded R-map models of optimal connectivity.

Estimating outcomes using the model: In direct parallel to the other two approaches, spatial similarities between single connectivity fingerprints and R-map models were calculated using voxel-wise spatial correlations. This led to positive high correlation values for cases in which fingerprints graphically matched the (optimal) connectivity profile represented by the R-map model – and lower or even negative values for the opposite cases. The R-model obtained by combining all the single connectivity fingerprints was cross-validated using a leave-one-out and multiple k-fold [$k$ = 3, 5, 7, 10] designs, and quantitatively and interactively compared to the Neurosynth database (neurosynth.org) to allow comparison of the identified map to functional networks previously reported by other studies.

Analyses on the three levels (fiber filtering, sweetspot mapping and functional network mapping) were repeated using absolute (instead of relative) improvements of ADAS-cog 11 following DBS (supplementary Fig. 8), as well as improvements measured by ADAS-cog 13 scores (supplementary Fig. 9). In the latter, only the subset of patients from the ADvance trial were included (since in other patients, ADAS-cog 13 improvements were not available).

### Analysis of flashback-like phenomena

During the surgical intervention of a subset of the cohort, 39 patients aged 67.7 ± 7 years old, 19 females (participants from ADvance fx-DBS trial[29,33], NCT01608061), it was tested whether flashback phenomena could be induced[30,31] by means of stimulation with increasing voltages (1–10) at multiple contacts (0–3), eliciting at least one memory flashback in 18 (8 females) of these patients. This resulted in a total of 2054 stimulation volume probes, 66 of which evoked a flashback-like episode. We investigated the presence of streamlines correlated to these stimulation volumes in the same fashion as we did for our whole dataset using the Fiber Filtering Tool (Fig. 1a).

### Reporting summary

Further information on research design is available in the Nature Portfolio Reporting Summary linked to this article.

## Data availability

Anonymized derivatives of stimulation data used for the described analyses are openly available on OSF (https://osf.io/bckuf). The resulting tract atlas, sweet spot and fMRI network pattern are openly available within Lead-DBS software (www.lead-dbs.org).

Normative data:

Structural connectome: https://datadryad.org/stash/dataset/doi:10.5061/dryad.nzs7h44q2

Functional connectome: https://dataverse.harvard.edu/dataset.xhtml?persistentId=doi:10.7910/DVN/25833

Neurosynth database: https://github.com/neurosynth/neurosynth-data.

## Code availability

All code used to analyze the dataset is openly available within Lead-DBS/-Connectome software (https://github.com/leaddbs/leaddbs). Code to reproduce main results and figures is openly available on OSF (https://osf.io/bckuf).

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

## Acknowledgements

The ADvance Study team included: **Functional Neuromodulation:** Todd Langevin, Lisa Fosdick, Kristen Drake, Donald E. Reymers, Robyn Moxon, Dan O'Connell, Vince Owens, Cara Pendergrass, Susan Klees, Steven D. Targum. and the seven participating clinical trial sites: **Chair's Office at Johns Hopkins University and University of Toronto**: Constantine G. Lyketsos, MD, MHS, Co-PI, Elizabeth Plank Althouse Professor and Chair of Psychiatry and Behavioral Sciences at Johns Hopkins Bayview; Andres M. Lozano, MD, PhD, FRCSC, FACS, Co-PI, Professor, and Chair of Neurosurgery, Tasker Chair of Functional Neurosurgery; Gwenn Smith, PhD, Imaging Core Director, Richman Family Professor of Psychiatry and Behavioral Sciences, Johns Hopkins University; Cynthia Munro, PhD, Neuropsychologist, Associate Professor of Psychiatry and Behavioral Sciences, Johns Hopkins University; Esther Oh, MD, Medical Monitor, Assistant Professor of Geriatric Medicine, Johns Hopkins University; Jeannie Sheppard Leoutsakos, PhD, Data Core Leader, Assistant Professor of Psychiatry and Behavioral Sciences, Johns Hopkins University. **Clinical Trial Sites**. *Banner Alzheimer's Institute, Phoenix*: Anna Burke, MD, Geriatric Psychiatrist, Dementia Specialist; Francisco A. Ponce, MD, Associate Professor of Neurosurgery, Director Barrow Center for Neuromodulation. *Banner Sun Health Research Institute, Sun City*: Marwan Sabbagh, MD, Director, Banner Sun Health Research Institute; Francisco A. Ponce, MD, Associate Professor of Neurosurgery, Director Barrow Center for Neuromodulation. *Brown University, Rhode Island Hospital, Butler Hospital*: Stephen Salloway, MD/MS Professor of Neurology, Director of Neurology and Memory and Aging Program; Rees Cosgrove, MD/PhD, Chair of Neurosurgery; Wael Asaad, MD/PhD, Assistant Professor of Neurosurgery. *Johns Hopkins University School of Medicine, Baltimore MD*: Paul Rosenberg, MD, Associate Professor, Associate Director, Memory and Alzheimer's Treatment Center; William S. Anderson, MD, PhD, Associate Professor of Neurosurgery, Zoltan Mari, M. D, Associate Professor of Neurology, Ned Sacktor, MD, Professor of Neurology. *University of Florida – Gainesville*: Michael S. Okun, MD, Professor of Neurology, Co-Director of the Center for Movement Disorders and Neurorestoration; Kelly D. Foote, MD, Professor of Neurosurgery, Co-Director for Center for Movement Disorders and Neurorestoration. *University of Pennsylvania*: David A. Wolk, MD, Associate Professor of Neurology, Assistant Director Penn Memory Center; Gordon Baltuch, MD/PhD, Professor of Neurosurgery, Director Center for Functional and Neurorestorative Neurosurgery. *University of Toronto/Toronto Western Hospital*: Andres M. Lozano, MD, PhD, FRCSC, FACS, Professor of Neurosurgery, Tasker Chair of Functional Neurosurgery; David F. Tang-Wai, MDCM FRCPC, Associate Professor of Neurology.

## Author contributions

A.S.R. and A.H. designed the study, analyzed the data, and wrote the manuscript. S.O., C.N., K.B., N.L. and N.R. analyzed the data and critically revised the manuscript, K.B. performed in-fold analysis. A.B., G.J.B.E., J.G., A.L., W.D., B.S., L.B.d.A., K.D.F., R.A., P.B.R., D.F.T.W., D.A.W., A.D.B., S.S., M.N.S., M.M.C., G.S.S., C.G.L., M.S.O., W.S.A., Z.M., F.A.P., A.M.L. collected the data and critically revised the manuscript. F.W. and K.S. provided the normative structural data and critically revised the manuscript.

## Funding

## Competing interests

C.N. was supported by the German Research Foundation (Deutsche Forschungsgemeinschaft, DFG NE 2276/1-1). K.F. received grants and personal fees from Medtronic and Boston Scientific, grants from Abbott/St. Jude, and Functional Neuromodulation outside the submitted work. D.W. received grants from Functional Neuromodulation during conduct of this study, grants and personal fees from Avid/Lily, and Merck, personal fees from Jannsen, GE Healthcare, Biogen and Neuronix outside the submitted work. S.S. receives personal fees from Elsai, Lilly, Roche Novartis and Biogen outside the submitted work. M.S. received personal fees from Allergan, Biogen, Roche-Genentech, Cortexyme, Bracket,

Sanofi, and other type of support from Brain Health Inc and uMethod Health outside of the submitted work. C.L. received grants from Functional Neuromodulation Inc. during conduct of this study, from Avanir and Eli Lily and NFL Benefits Office outside of the submitted work. M.O. received grants from NIH, Tourette Association of America Grant, Parkinson's Alliance, Smallwood Foundation, and personal fees from Parkinson's Foundation Medical Director, Books4Patients, American Academy of Neurology, Peerview, WebMD/Medscape, Mededicus, Movement Disorders Society, Taylor and Francis, Demos, Robert Rose and non-financial support from Medtronic outside of the submitted work. A.L. received grants from Medtronic and Functional Neuromodulation during conduct of this study, personal fees from Medtronic, St. Jude, Boston Scientific, and Functional Neuromodulation outside of submitted work. A.L. disclosed having a patent "US Patent 8,346,365. Lozano AM. Cognitive function within a human brain. 2013" licensed to Functional Neuromodulation. A.H. was supported by the German Research Foundation (Deutsche Forschungsgemeinschaft, 424778381 – TRR 295), Deutsches Zentrum für Luft- und Raumfahrt (DynaSti grant within the EU Joint Programme Neurodegenerative Disease Research, JPND), the National Institutes of Health (R01 13478451, 1R01NS127892-01 & 2R01 MH113929) as well as the New Venture Fund (FFOR Seed Grant). ADvance was supported by the National Institute on Aging (R01AG042165) and Functional Neuromodulation Ltd., the sponsor of the study. Other co-authors report no conflicts of interest.

## Additional information

[1]Movement Disorder and Neuromodulation Unit, Department of Neurology, Charité – Universitätsmedizin Berlin, corporate member of Freie Universität Berlin and Humboldt-Universität zu Berlin, Berlin, Germany. [2]Division of Neurosurgery, Department of Surgery, University Health Network and University of Toronto, Toronto, ON M5T2S8, Canada. [3]Krembil Research Institute, University of Toronto, Toronto, ON M5T2S8, Canada. [4]Joint Department of Medical Imaging, University of Toronto, Toronto, ON M5T1W7, Canada. [5]UMass Chan Medical School, Department of Neurology, Worcester, MA 01655, USA. [6]UMass Memorial Health, Department of Neurology, Worcester, MA 01655, USA. [7]Athinoula A. Martinos Center for Biomedical Imaging, Department of Radiology, Harvard Medical School, Massachusetts General Hospital, Charlestown, MA, USA. [8]Harvard-MIT Health Sciences and Technology, MIT, Cambridge, MA, USA. [9]Department of Radiology, Stanford University, Stanford, CA, USA. [10]University of Florida Health Jacksonville, Jacksonville, FL, USA. [11]Norman Fixel Institute for Neurological Diseases, Departments of Neurology and Neurosurgery, University of Florida, Gainesville, FL, USA. [12]Cerebral Imaging Centre, Douglas Research Centre, Montreal, QC, Canada. [13]Department of Psychiatry and Behavioral Sciences and Richman Family Precision Medicine Center of Excellence, School of Medicine, Johns Hopkins University, Baltimore, MD, USA. [14]Department of Medicine, Division of Neurology, University Health Network and University of Toronto, Toronto, ON M5T2S8, Canada. [15]Department of Neurology, University of Pennsylvania, Philadelphia, PA, USA. [16]Barrow Neurological Institute, Phoenix, AZ, USA. [17]Department of Psychiatry and Human Behavior and Neurology, Alpert Medical School of Brown University, Providence, RI, USA. [18]Memory & Aging Program, Butler Hospital, Providence, USA. [19]Department of Psychiatry, McGill University, Montreal, QC, Canada. [20]Biological and Biomedical Engineering, McGill University, Montreal, QC, Canada. [21]Johns Hopkins School of Medicine, Baltimore, MD, USA. [22]Cleveland Clinic Lou Ruvo Center for Brain Health, Las Vegas, NV, USA. [23]Center for Brain Circuit Therapeutics, Department of Neurology, Brigham and Women's Hospital, Boston, MA, USA. [24]Departments of Neurology and Neurosurgery, Massachusetts General Hospital, Boston, MA, USA. ✉e-mail: ahorn1@bwh.harvard.edu

