## [Peer Review File · Nature Communications]

Optimal Stimulation Sites and Networks for Deep Brain Stimulation of the Fornix in Alzheimer's DiseaseReviewers' comments:

Reviewer #1 (Remarks to the Author):

In this study, Ríos et al. examine which electrode locations for deep brain stimulation of the fornix lead to beneficial cognitive effects in Alzheimer patients. In particular, the authors performed a post hoc analysis on a series of patients from the phase I trial by Laxton et al 2010, the phase II trial (ADvance trial) by Lozano et al 2016 and an on-going trial on biomarkers and dose optimization of fornix DBS (NCT04856072). In these patients, the variability in DBS electrode placement was investigated on three levels: i) effects of focal electric fields of stimulation on white matter tracts traversing the stimulation volumes ii) optimal stimulation sites on a localized voxel level, and iii) impact of fornix DBS on distributed whole-brain functional networks. The authors have validated their electrode localization method and subsequent DBS fiber filtering and network mapping approaches in an impressive and convincing set of previous publications (for example Horn et al., Neuroimage 2019; Baldermann et al., Biol Psych 2019).

Overall, this study provides a very interesting and promising tool to define the most optimal stimulation target of fornix DBS in AD. Despite elegant sets of experiments and the application of a very sophisticated technique, some questions remain open.

Specifically:

The majority of patients used in the present study stem from the ADvance trial (N= 40). What can be said about the ADvance trial retrospectively? The conclusion of the trial was that only patients over 65 years seem to benefit while there was possible worsening in patients below age 65 years with stimulation. Can the findings of the current study shed light on why only “older” patients derived benefit? Was there a difference in electrode locations? I miss a more thorough discussion specifically with regard to the conclusions of previous trials.

Another conclusion of the ADvance trial was that the stimulation parameters applied to AD patients were not disease-specific. Developing AD-specific stimulation parameters is also likely to improve the current approach of DBS in AD. The authors do not discuss stimulation parameters at all in the present manuscript. Can the authors speculate which stimulation parameters of their target location would yield most optimal effects in patients?

Minor comments:

1) Most figures are not very clear. For example the asterisk in Figure 3 is hard to see. In Figure 4, A and B are missing in the figure and in B axes and legends are not legible. Figure 5, yellow is hard to see.

2) P.13 line 312 sentence is not complete.

3) The ADvance trial included 42 patients, but only 40 patients were used in the current study. Why were 2 patients excluded?

Reviewer #2 (Remarks to the Author):

This paper presents a well-researched work on investigating the effect of fornix-DBS in AD. The authors perform extensive cross-validation of their proposed method on a large cohort of 50 subjects. Normative atlas based tractogram is used to discern the tracts that provide positive or negative outcomes. Further, functional connectivity related to the outcome is also shown. Map of the sweet and sour spots

For stimulation are presented. One drawback seems to be that each individual subject T1/T2 data is

registered to a healthy brain atlas — which could create inaccuracies in the localization as acknowledged by the authors.

For example, if the fornix has significantly atrophied in an AD subject, how accurate is the localization of this anatomy to a healthy subject? — perhaps this point could be emphasized in the limitations a bit further.

But otherwise, a nice piece of work.

Comments:

1. Figure 2 caption and text: Poor responders are said to be those with increase in ADAS-cog-11 score — whereas in the plot, poor responders are ones with decrease in ADAS-cog-11 scores. Please fix the caption and the relevant text.
2. The peak or maximum E-field based on which the fiber scores are calculated are by nature noisy (since the max values can vary based on model parameters). Any reason why only the max E-field value was used and not the “capture of the fiber population stimulated” as depicted by the red and blue “blobs” in Figure 2B and 2C ?
3. Line 217: The use of the word test-retest seems inappropriate here - as it is typically used when the same set of subject data is acquired and tested twice or more number of times. Recommend using “separate validation” or equivalent word.
4. The software where Wrapdrive was implemented and made available should be provided.
5. Line 562: Not sure where or how the number 103-105 axons per fiber bundle was arrived at. Most axons in the human brain are less than 5 um with an overwhelming majority being around 2 um or less. So a cubic millimeter could have many more than 103-105 axons.
6. Line 588: Is the E-field vector or the peak magnitude of the E-field vector that was correlated with outcome? If it is not the magnitude, please explain how the vector was used in the correlation.
7. Did the stimulation parameters vary during the year ? i.e. were adjustments made to the parameters for each subject. If so, which setting was used in the analysis and why? This would determine the peak E-field and thus the fiber-score as well as the selected fibers. Please clarify.
8. The “optimal predictor fibers” seem to be looping in Figure 2 — is that realistic anatomy? If so, please provide some reference on the existence of such a tract.

Minor:

Line 194: BNST: acronym used without definition

Line 585: MIO : acronym not defined

Reviewer #3 (Remarks to the Author):

Summary: In this work, the authors curated a multi-site data set comprised of 50 patients with mild Alzheimer’s Disease that underwent deep brain stimulation. The aim of their study is to understand the neural substrates associated with successful fornix DBS, where success is defined as a clinical improvement (measured as improvement in cognitive scores one year after DBS treatment). Their analyses were conducted at three levels: local, structural/tract, and functional connectivity. The

authors conclude that their results “propose a potential optimal stimulation target for Alzheimer’s Disease treatment with fx-DBS.”

Strengths: The authors tested multiple prediction models using different cross validation approaches and the results appear to be robust across these different approaches. They use advanced, state-of-the-art methods for the modeling of the DBS data and also use a tool (that included manual edits) for proper normalization of the data given the atrophy. Their methods for preparing the data have convinced me that this is high-quality data being fed into the predictive models and this helps with my concerns regarding the small sample size for predictive modeling (while N=50 might be considered large for a DBS study, it is considered a very small sample size for machine learning). Their research question is framed well, and enough clinical background is given for non-clinical audience to understand why this study is interesting and important.

Weaknesses: I am unable to properly review this paper as I am not a clinician, neuroanatomist, or DBS expert. It seems that reviewers from these backgrounds have been provided enough details to review the work. I was asked by the editors to review this work given my expertise in machine learning, connectivity and neuroimaging applications of ML, however, from my perspective (reviewing the predictive modeling aspects of the paper), there are simply not enough details provided to assess if any modeling mistakes or statistical violations have been made. I found that overall it was difficult to follow and understand their analysis and the input data. There is essentially no detail about how the predictive model was formulated, meaning there are no equations, descriptions of software tools used for statistical modeling, or any analysis code shared. After reading the paper (and supplemental files) three times, I cannot tell what the input data was to the predictive model. It is unclear if individual resting state data from each subject was used or if the functional data was used from some other study. Functional data from the AD patient groups is not described, and supplemental files state only that 1.5T structural MRI data was acquired. There is reference to 1,000 healthy subjects functional data, “normative connectome” and “connectivity fingerprints” but it is never described what these are and how these data were generated. There is also a description of a DWI dataset acquired from a single subject, but no description of the DWI data from each AD patient. It seems that the only MRI data acquired from each subject was the T1w/T2w volumes and no DWI or fMRI data. This is very misleading as the narrative makes it seem as if there is DWI and fMRI from each subject that went into the predictive model. If there is individual level DWI and fMRI data, these data need to be described (acquisition parameters, preprocessing methods, quality checking). The authors make claims regarding generalizability and robustness, however these claims are not supported by the statistical framework presented to the reader. Given the strong clinical claims (“Potentially, our results can be useful to guide DBS programming in existing patients with fx-DBS and potentially inform surgical targeting in AD within future investigational trials”), this lack of detail is very concerning and needs to be addressed before consideration of publication in any journal. Without much to go off of in terms of evaluating the machine learning framework, my main concerns are regarding the lack of model comparison across all of the test sets and different cross validation strategies and the lack of clarity about what data each subject contributed to the predictive model. Several statements made throughout the paper are concerning to me and are suggestive that proper statistical inference has not been performed and therefore the analysis/results do not support the conclusions of the paper. “Purely visual test-retest comparison of results”

No permutation testing or any null hypothesis testing

Only reporting cross validated Pearson correlation and p-values that are averaged across folds. There are no error bars/confidence intervals, or accuracy evaluation metrics (i.e., mean squared error, median absolute error) reported. This is a huge red flag

(<https://www.sciencedirect.com/science/article/pii/S1053811917305311> & <https://jamanetwork.com/journals/jamapsychiatry/article-abstract/2756204>)

General remark to all reviewers:

After discussions with the editor and all co-authors, we decided to exclude four patients from an on-going trial on biomarkers and dose optimization that were originally included in the first version of the manuscript. While the present study could be considered completely independent to the ongoing trial (NCT04856072), we mutually agreed on the necessity to avoid conflicts with reporting & design of the ongoing trial. Hence, the present study retrospectively included 6 patients from the phase I trial (Laxton et al. 2010, NCT00658125) and 40 patients from the phase II trial (ADvance trial, NCT01608061).

Results and all key conclusions remained largely unchanged and significant.

Reviewer #1 (Remarks to the Author):

In this study, Ríos et al. examine which electrode locations for deep brain stimulation of the fornix lead to beneficial cognitive effects in Alzheimer patients. In particular, the authors performed a post hoc analysis on a series of patients from the phase I trial by Laxton et al 2010, the phase II trial (ADvance trial) by Lozano et al 2016 and an on-going trial on biomarkers and dose optimization of fornix DBS (NCT04856072). In these patients, the variability in DBS electrode placement was investigated on three levels: i) effects of focal electric fields of stimulation on white matter tracts traversing the stimulation volumes ii) optimal stimulation sites on a localized voxel level, and iii) impact of fornix DBS on distributed whole-brain functional networks. The authors have validated their electrode localization method and subsequent DBS fiber filtering and network mapping approaches in an impressive and convincing set of previous publications (for example Horn et al., Neuroimage 2019; Baldermann et al., Biol Psych 2019).

Overall, this study provides a very interesting and promising tool to define the most optimal stimulation target of fornix DBS in AD. Despite elegant sets of experiments and the application of a very sophisticated technique, some questions remain open.

We would like to thank the reviewer for their overall very positive evaluation of our manuscript.

Specifically:

The majority of patients used in the present study stem from the ADvance trial (N= 40). What can be said about the ADvance trial retrospectively? The conclusion of the trial was that only patients over 65 years seem to benefit while there was possible worsening in patients below age 65 years with stimulation. Can the findings of the current study shed light on why only “older” patients derived benefit? Was there a difference in electrode locations? I miss a more thorough discussion specifically with regard to the conclusions of previous trials.

We would like to thank the reviewer for raising this very important point, while reiterating that in the Advance trial, this difference was shown post-hoc and only in one arm of the study. We have now amended multiple strands of analyses with the aim to compare i) stimulation sites and ii) response tracts between ages with age below and above 65 years, separately. Results did not show clear effects of age regarding these factors, which we now discuss in light of results from the ADvance trial:

“Age has emerged as a possible treatment effect modifier in the ADvance trial. Here, among individuals in the early-on arm during phase 1 (but not in phase 2), participants below the age of 65 worsened on the ADAS-cog13 significantly more than older participants³³.” – Introduction, p. 5

“Effects of age

Prior results had shown differences in clinical improvements related to age groups, where among individuals in the early-on arm during phase 1 (but not in phase 2), participants below the age of 65 worsened on the ADAS-cog13 significantly more than older participants, while those showed improvement³³. The robustness of models in the present study to successfully cross-estimate clinical improvements across the entire group regardless of age (and regardless of slicing up the data into leave-one-out, 10-, 7-, 5- and 3-fold cross-validation designs) does not a priori confirm such an effect (i.e., the same model seemed to be predictive in both age groups). An alternate reason for age differences could be (potentially atrophy related) systematic shifts in electrode placements. However, as can be seen in figure S10, no apparent difference in electrode placements was observed between the groups, if at all more variability on the z-axis in the young cohort. Furthermore, there was no significant difference in fiber scores obtained across the two age groups ($p = 0.790$). This does not suggest a systematic shift between groups (such as stimulation in younger participants systematically modulating optimal fiber connections less strongly than in older participants).” – results, p. 15

“In this regard, we were not able to find apparent differences in i) electrode placement or ii) fiber-score activations between patients younger than versus older than 65, which suggests other factors might have influenced the clinical outcome in the younger group (Figure S10). As previously reported, possible explanations for the decline in early onset subjects include a more aggressive presentation of the condition, greater brain atrophy and comparably more reduced glucose metabolism in this subgroup of patients^{29,60,66,67}.” – discussion, p. 21

Figure S10. Effects of Age. A) Axial, coronal, and sagittal overlay of maps created from stimulation volumes of subjects older than 65 years (left), younger than 65 years (middle) and whole cohort (right). B) Fiberscores obtained through DBS fiber filtering analysis explained in Methods and Results sections, by the stimulation volumes of younger than 65-year-old patients (top), and patients 65-year-old or older (bottom), $p(T\text{-test}) = 0.790$. The model used to estimate these scores was calculated in a leave-one-patient out design across the entire cohort. - Supplementary Information

Another conclusion of the ADvance trial was that the stimulation parameters applied to AD patients were not disease-specific. Developing AD-specific stimulation parameters is also likely to improve the current approach of DBS in AD. The authors do not discuss stimulation parameters at all in the present manuscript. Can the authors speculate which stimulation parameters of their target location would yield most optimal effects in patients?

Unfortunately, the retrospective nature of our study prevents us from analyzing different stimulation paradigms (all patients received continuous stimulation of 130 Hz frequency with pulse-widths of 90 ms). Then, stimulation parameters (frequency and pulse width) would not have a clear effect on the stimulation volumes since these are analyzed in static fashion while frequency and pulse width will likely have effects predominantly on the temporal domain. Modeling this in the fornix region is complex, since different bundles of question (e.g., fornix itself, stria terminalis, etc.) have different (in part unknown) axonal properties.

So stated differently, while our results may add clarity to the question of **where** to stimulate, much less if nothing can be said about **how** and **when** to stimulate. The following paragraph was added to the limitations section to clarify this:

“The retrospective nature of our study prevented us from analyzing different effects of stimulation frequencies, pulse widths, or stimulation patterns, which would enfold different signals onto the network over time. Instead, the imaging nature of our study analyzes results in static fashion (both on a stimulation volume and network level). Future research is needed to investigate effects of variations in stimulation parameters, such as the ongoing trial to optimize electrical stimulation parameters of fornix-DBS for AD (NCT04856072). Alternatively, neuromodulation delivered through distinct approaches, namely, the ongoing trial on gamma entrainment via sensory stimulus at a 40Hz frequency (NCT04055376) could extend our knowledge on the effect of diverse parameters in brain stimulation for AD.” – discussion, p. 20

Minor comments:

1) Most figures are not very clear. For example, the asterisk in Figure 3 is hard to see. In Figure 4, A and B are missing in the figure and in B axes and legends are not legible. Figure 5, yellow is hard to see.

All mentioned figures were revised for clarity. For figure 5 and related figures (2, S8, S9), we now decided to not color-code by improvement, because it was redundant information, and the groups (top, middle, poor responders) did not inform our statistical analysis in any way. The revised versions of mentioned figures are pasted below:

Figure 3. Probabilistic mapping of sweet and sour spots associated with clinical outcome. A) Identified clusters of sweet (red) and sour (blue)-spots in a 3D view, superimposed on slices of a 100- μ m, 7T brain scan in MNI 152 space⁴⁸. Since the result was symmetric, on the bottom of the panel, we flipped stimulation volumes across hemispheres to further increase robustness on a voxel-level (effectively doubling the number of electrodes used in each hemisphere). B) Axial, coronal, and sagittal views of sweet and sourspot peak coordinates (also see table S6). Projections of cluster center coordinates are marked by a black asterisk and directly project onto the intersection between fornix and bed nucleus of stria terminalis (BNST, see also Fig. S6). C) Axial, coronal, and sagittal sections showing DBS fiber filtering results obtained from the whole cohort at MNI: $X = -3.6$, $Y = -1.5$, and $Z = -3.6$. Abbr: Put: Putamen, Cdt: Caudate, ALIC: Anterior limb of the internal capsule, AC: Anterior commissure, GPe/i: external / internal pallidum, Thal: thalamus, RN: red nucleus, MB: mamillary bodies, Fx: Fornix. Fornix is shown in blue-green color, informed by the CoBrALab Atlas⁴⁷. Bed nucleus of the stria terminalis shown in light brown color, informed by Neudorfer et al.³⁹.

Figure 4: A) Functional networks associated with optimal improvements across training (left), test (middle) and combined (right) cohorts. Brain regions are color-coded by correlations between degree of functional connectivity with DBS electrodes and clinical improvements across the cohorts. Since results were highly symmetric, only the left hemisphere is shown. B) Optimal network associations to Neurosynth database terms, left: highlighted relevant regions for the most similar networks identified; right: similarity plots between same networks and optimal network identified by Network Mapping results (x-axis = specific network meta-analysis, z-score, y-axis = DBS Network Map).

Figure 5. Results summary including the models from DBS fiber filtering, sweetspot mapping and network-mapping. The three levels of analysis were able to explain a similar amount of variance of clinical outcomes when analyzed in a circular nature (see scatterplots; ~16-19%) and led to significant cross-predictions of clinical outcomes across leave-one-patient-out and multiple k-fold designs. Three level analysis results were superimposed on slices of a brain cytoarchitecture atlas in MNI 152 space⁵². Gray shaded areas represent 95% confidence intervals, see figure S7 for additional metrics on each validation approach.

2) P.13 line 312 sentence is not complete.

This was fixed, the sentence now reads:

“To allow a certain 299 degree of reverse inference of these network results⁵⁰, they were spatially compared to maps 300 associated with a total of 1307 terms present in the Neurosynth database 301 (<https://neurosynth.org/>)⁵¹” –Results, p. 13

3) The ADvance trial included 42 patients, but only 40 patients were used in the current study. Why were 2 patients excluded?

Indeed, for the missing 2 patients, imaging data to reconstruct DBS electrodes was unavailable (as was the case in previous imaging studies, e.g. Neudorfer et al. Brain 2021, Germann et al. Alzheimers Dement. 2020). This was clarified:

“While the ADvance trial included 42 patients, imaging data was only available for 40 patients^{31,32} (also see figure S1).” – Methods, p. 23

Figure S1. Flowchart explaining patient inclusion for this work.

Reviewer #2 (Remarks to the Author):

This paper presents a well-researched work on investigating the effect of fornix-DBS in AD. The authors perform extensive cross-validation of their proposed method on a large cohort of 50 subjects.

Normative atlas based tractogram is used to discern the tracts that provide positive or negative outcomes. Further, functional connectivity related to the outcome is also shown.

Map of the sweet and sour spots

For stimulation are presented. One drawback seems to be that each individual subject T1/T2 data is registered to a healthy brain atlas — which could create inaccuracies in the localization as acknowledged by the authors.

For example, if the fornix has significantly atrophied in an AD subject, how accurate is the localization of this anatomy to a healthy subject? — perhaps this point could be emphasized in the limitations a bit further.

We would like to thank the reviewer for their overall very positive evaluation of our manuscript. Indeed, the registration includes bias but allowed the tract analysis, in the first place (patients need to be registered to a joint space for the fiber filtering concept to work, by design). In fact, we had failed in creating registrations in this cohort that met our satisfaction/quality standards in the past (as opposed to e.g., cohorts with Parkinson's Disease or OCD). Only the development of the WarpDrive tool allowed us to carry out this study. We would like to point the reviewer to Video S1 which demonstrates this approach. In our view, WarpDrive allowed us to accurately register the fornix regions satisfactorily, which was a key "ingredient" and innovation allowing us to carry out this study.

We still agree registration inaccuracy constitutes a potential source of bias and have further emphasized the discussion of this point in our limitations section:

"For the present study, this was crucial, since in the field of DBS, electrode displacements of a few millimeters will lead to substantially different effects^{35,37}. To account for this, we applied a novel method, termed WarpDrive⁶⁷, to manually refine registrations into template space (see video S1). Briefly, WarpDrive provides a graphical interface allowing precise alignment of source and target landmarks by directly visualizing the normalized images, together with the template and atlases in MNI space. WarpDrive allows to correct for misalignments and recomputes a refined deformation field in real time." — Methods, p. 24

But otherwise, a nice piece of work.

Thank you!

Comments:

1. Figure 2 caption and text: Poor responders are said to be those with increase in ADAS-cog-11 score — whereas in the plot, poor responders are ones with decrease in ADAS-cog-11 scores.

Please fix the caption and the relevant text.

We agree and flipped the signs accordingly also for the % improvement (had been calculated as (pre-post)/pre, now is (post-pre)/pre, resulting in a sign flip) as well as showed values for each patient in novel table S3 (along with the equation that was used to calculate them):

Table S3. ADAS-cog 11 scores and group assignment of patients. Absolute change calculated subtracting Baseline ADAS-cog 11 from 12-month ADAS-cog 11 value.

Patient ID	Baseline ADAS-cog 11	12-month ADAS-cog 11	Absolute change (post-pre)	Relative change (post-pre)/pre	Group
01	28	34	6	21.42	Poor responders
02	22	30	8	36.36	Poor responders
03	19	34	5	78.94	Poor responders
04	17	39	22	129.41	Poor responders
05	19	21	2	10.52	Middle responders
06	13	18	5	38.46	Poor responders
07	13	15	2	15.38	Middle responders
08	24	31	7	29.16	Poor responders
09	23	30	7	30.43	Poor responders
10	13	24	11	84.61	Poor responders
11	12	7	-5	-41.67	Top responders
12	15	24	9	60	Poor responders
13	31	36	5	16.13	Middle responders
14	29	43	14	48.28	Poor responders
15	19	26	7	36.84	Poor responders
16	32	33	1	3.13	Middle responders
17	16	29	13	81.25	Poor responders
18	18	23	5	27.78	Poor responders

Patient ID	Baseline ADAS-cog 11	12-month ADAS-cog 11	Absolute change (post-pre)	Relative change (post-pre)/pre	Group
19	23	24	1	4.35	Middle responders
20	15	36	11	140	Poor responders
21	22	10	-12	-54.55	Top responders
22	16	19	3	18.75	Middle responders
23	16	22	6	37.5	Poor responders
24	21	42	21	100	Poor responders
25	17	30	13	76.48	Poor responders
26	24	29	5	20.83	Middle responders
27	28	38	10	35.71	Poor responders
28	14	15	1	7.14	Middle responders
29	20	28	8	40	Poor responders
30	16	15	-1	-6.25	Top responders
31	17	12	-5	-29.41	Top responders
32	35	51	16	45.71	Poor responders
33	22	39	17	72.27	Poor responders
34	21	23	2	9.52	Middle responders
35	17	35	18	105.88	Poor responders
36	22	18	-4	-18.18	Top responders
37	19	19	0	0	Middle responders
38	18	17	-1	-5.56	Top responders
39	16	15	-1	-6.25	Top responders
40	13	21	8	61.54	Poor responders
41	18	17	-1	-5.56	Top responders
42	11	17	6	54.55	Poor responders
43	21	22	1	4.76	Middle responders
44	13	40	27	207.69	Poor responders
45	10	19	9	90	Poor responders
46	22	19	-3	-13.64	Top responders

2. The peak or maximum E-field based on which the fiber scores are calculated are by nature noisy (since the max values can vary based on model parameters). Any reason why only the max E-field value was used and not the “capture of the fiber population stimulated” as depicted by the red and blue “blobs” in Figure 2B and 2C?

We apologize this was unclear, indeed we did the latter (populations of tracts were used to calculate scores). However, each E-field intersects with each tract on multiple points along the tract. From these points, only the peak was used (). The rationale is that the recruitment of the fiber is defined by the local perturbation of the extracellular field, hence the higher the peak of the E-field magnitude, the more likely is the recruitment by the stimulation. And while the cable model defines the fiber activation by the second derivative of the extracellular potential, the rationale is appropriate for conventional DBS stimulations, where the electric field distribution is trivial.*

This was further clarified:

“Model definition (Figure 1A): Whole brain structural connectivity profiles seeding from bilateral E-fields were calculated using a state-of-the-art multi-shell diffusion-weighted imaging dataset acquired across 18 scanning hours of a single individual at 760 μm isotropic resolution⁴¹ using the generalized q-sampling approach (default parameters) and whole-brain tracking (default parameters) as implemented in DSI studio⁸⁴. The patients were distributed into two cohorts: Training (N = 28) and Test (N = 18). For each subject of the training cohort, fibers traversing each voxel of the E-field were selected from the 5 million tracts in the normative connectome and projected to a voxelized volume in MNI space. Each of these fibers were weighted according to the E-field magnitude at each voxel, considering only fibers that traversed > 20% of stimulation volumes with an E-field magnitude > 0.36 V/mm. Each fiber was then appointed an R-value dependent on the Spearman correlation between its weighting and the respective clinical outcome scores across the group, i.e., a high R-value indicates that the modulation of that tract is associated with clinical improvement. Given the mass-univariate nature of this approach (and subsequent alpha-error accumulations), the resulting correlation coefficients were not considered significant, but were rather used to discriminate and visualize a specific set of bundles that was later validated by estimating clinical outcome in out of sample data (Figure 1, table S5).” – Methods, page 25-26.

(As an aside, the Lead-DBS software also includes the function to use the 5% top peak points on each tract – we tested this and results were unchanged.*

3. Line 217: The use of the word test-retest seems inappropriate here - as it is typically used when the same set of subject data is acquired and tested twice or more number of times. Recommend using “separate validation” or equivalent word.

This was changed to “separate validation”, throughout the revised manuscript.

4. The software where Warpdrive was implemented and made available should be provided.

Warpdrive is available as a module in 3DSlicer (Add package -> SlicerNetstim) and available as open source here: <https://github.com/netstim/SlicerNetstim>. It is also included in Lead-DBS when using the develop branch openly available when using `github` to `install` Lead-DBS: <https://netstim.gitbook.io/leaddbs/installation#installation-via-github> Finally, WarpDrive is explained to some degree here: <https://elifesciences.org/articles/72929> and has been presented as a posted at multiple

conferences (e.g. http://www.netstim.org/wp-content/uploads/2022/03/DBSExpertSummit_Poster_Oxenford.pdf, additional demo video available at <https://youtu.be/VcBXu5BURVI>).

A link to the github code for WarpDrive was added to the manuscript:

“Briefly, WarpDrive provides a graphical interface allowing precise alignment of source and target landmarks by directly visualizing the normalized images, together with the template and atlases in MNI space (the software is openly available here: <https://github.com/netstim/SlicerNetstim>). WarpDrive allows the user to manually correct misalignments from the standard normalization and recomputes a refined deformation field in real time.” – Methods, p. 24

5. Line 562: Not sure where or how the number 103-105 axons per fiber bundle was arrived at.

Most axons in the human brain are less than 5 um with an overwhelming majority being around 2 um or less.

So a cubic millimeter could have many more than 103-105 axons.

*We are sorry for this formatting error; the sentence was supposed to state 10^3 - 10^5 axons. This was corrected and the reference (Zalesky, A. & Fornito, A. A DTI-derived measure of cortico-cortical connectivity. *IEEE Trans Med Imaging* **28**, 1023–1036 (2009).) was added to the correct place in the sentence.*

6. Line 588: Is the E-field vector or the peak magnitude of the E-field vector that was correlated with outcome?

If it is not the magnitude, please explain how the vector was used in the correlation.

The E-field magnitude was used; this was made clear as follows:

“Each of these fibers were weighted according to the E-field magnitude at each voxel, considering only fibers that traversed > 20% of voxels with an E-field magnitude > 0.36 V/mm and Spearman rank correlated to the respective clinical outcome scores across the group.” – Methods, p. 26

7. Did the stimulation parameters vary during the year? i.e. were adjustments made to the parameters for each subject. If so, which setting was used in the analysis and why?

This would determine the peak E-field and thus the fiber-score as well as the selected fibers. Please clarify.

Stimulation parameters did not vary during the year. 12 month stim data was used across all patients building on the ADvance trial data:

“Patients were diagnosed by standardized criteria after expert examination rated with 0.5 or 1 on the Clinical Dementia Rating scale (CDR) and scored 12-24 on the Alzheimer’s Disease Assessment Scale 11 (ADAS-cog)⁷², further inclusion and exclusion criteria for the trials can be found in supplementary tables S1 and S2, patients received monopolar stimulation at a frequency of 130 Hertz with a 90 millisecond pulse width for 12 months without adjustment.” – Methods, p. 23

8. The “optimal predictor fibers” seem to be looping in Figure 2 — is that realistic anatomy? If so, please provide some reference on the existence of such a tract.

The perspective gives the illusion of a closed loop, however, this is a conflation of fornix and anterior commissure fibers. We have revised figure S4 to include other viewpoint

angles that debunk this illusion. The following discussion sections were added to prevent readers from perceiving the results:

“Lateral and top views of fibertract superimposed with structures of interest, white arrow indicates intersection of streamlines of the fornix and AC that could give the illusion of a loop on lateral projection views.” – legend to figure S4 (see below)

“Predictive fibers calculated on training and test cohorts alone were remarkably similar, each suggesting a strong involvement of fornix, anterior nuclei of the thalamus and stria terminalis, the anterior commissure was involved only when analyzing the training cohort, and the combined cohort.” – Discussion, p. 18

Figure S4. Fiber tracts associated with optimal clinical response superimposed on slices of a 100- μ m, 7T brain scan in MNI 152 space. From a set of 5 million fiber tracts sampled from a high-resolution connectome, the ones preferably modulated by top-responding (and not by poor-responding) patients were selected using the DBS fiber filtering method and visualized. The process was repeated on the training-cohort ($N = 30$) (A), the test-cohort ($N = 20$) (B), and both cohorts combined ($N = 50$) (C). Fiber tracts are color-coded by the resulting Spearman’s rank correlation coefficients which shows how strongly modulating each bundle correlated with clinical response across patients. D) Results from panel C superimposed on atlas structures forming part of the circuit of Papez, also visualized by dotted arrows. E) Lateral and top views of fibertract superimposed with structures of interest, white arrow indicates intersection of streamlines of the fornix and AC that could give the illusion of a loop on lateral projection views. 1. Hipp = Hippocampus, 2. Fx. = Fornix, 3. MB = mammillary bodies, 4. MMT = mamillothalamic

tract, 5. Thal. = thalamus, 6. Cg Cingulate gyrus, 7. Cingulum and 8. Parahipp: Parahippocampal gyrus. The backdrop features an ultra-high resolution (100 μm) template of the human brain⁶. Structures: Fornix (blue-green), Hippocampus (pink), Thalamus(blue) informed by the CoBrALab Atlas⁵, Bed nucleus of the stria terminalis (light brown) informed by the Atlas of the Human Hypothalamus⁷.

Minor:

Line 194: BNST: acronym used without definition

Line 585: MIO : acronym not defined

Thank you, both were clarified (Bed nucleus of the Stria Terminalis; Millions).

Reviewer #3 (Remarks to the Author):

Summary: In this work, the authors curated a multi-site data set comprised of 50 patients with mild Alzheimer's Disease that underwent deep brain stimulation. The aim of their study is to understand the neural substrates associated with successful fornix DBS, where success is defined as a clinical improvement (measured as improvement in cognitive scores one year after DBS treatment). Their analyses were conducted at three levels: local, structural/tract, and functional connectivity. The authors conclude that their results "propose a potential optimal stimulation target for Alzheimer's Disease treatment with fx-DBS."

We would like to thank the reviewer for their thorough evaluation of our work. We apologize, that when seen from a machine-learning standpoint, several points had been written up too vaguely (written with a medical audience/readership in mind).

We now i) share code and anonymized data to reproduce the presented plots (<https://osf.io/bckuf>) and ii) added a multitude of additional analyses after conferring with Russ Poldrack and Gaël Varoquax. Our study may not fulfill all criteria imposed by them in their excellent JAMA article (especially the N of the patient cohort, while representing almost all patients available, world-wide, to date, does not fulfill their community standards). However, their guidelines were developed for the field of psychology / neuroimaging and may not be 100% transferable to DBS. In fact, our study is the first in the field of DBS to show i) training -> test predictions in DBS for Alzheimer's and ii) a multitude of levels of cross-validations. Beyond sharing all data & a portable code that reproduces all data figures, we now add additional sub-analyses and report metrics that may further clarify our approach and robustness of results.

Strengths: The authors tested multiple prediction models using different cross validation approaches and the results appear to be robust across these different approaches. They use advanced, state-of-the-art methods for the modeling of the DBS data and also use a tool (that included manual edits) for proper normalization of the data given the atrophy. Their methods for preparing the data have convinced me that this is high-quality data being fed into the predictive models and this helps with my concerns regarding the small sample size for predictive modeling (while N=50 might be considered large for a DBS study, it is considered a very small sample size for machine learning). Their research question is framed well, and enough clinical background is given for non-clinical audience to understand why this study is interesting and important.

We would like to thank the reviewer for highlighting these positive points. We agree that the N of the study is a limitation, hence the multiple layers of cross-validation given the potential clinical importance of our study. The models we assume are linear, so it could

be seen as a study using classical statistics (with out-of-sample estimates) rather than a proper machine learning study (which may often use SVMs, neural networks or similar).

Weaknesses: I am unable to properly review this paper as I am not a clinician, neuroanatomist, or DBS expert. It seems that reviewers from these backgrounds have been provided enough details to review the work. I was asked by the editors to review this work given my expertise in machine learning, connectivity and neuroimaging applications of ML, however, from my perspective (reviewing the predictive modeling aspects of the paper), there are simply not enough details provided to assess if any modeling mistakes or statistical violations have been made. I found that overall it was difficult to follow and understand their analysis and the input data. There is essentially no detail about how the predictive model was formulated, meaning there are no equations, descriptions of software tools used for statistical modeling, or any analysis code shared.

*We would like to apologize again. Code & anonymized data are now transparently shared (<https://osf.io/bckuf>). The repository includes scripts with the statistical analyses and regenerate plots from figures 2 and 5 as well as some novel supplementary figures and should work out-of-the-box (without dependencies) using Matlab. The models are simple linear models based on a single regressor. However, this regressor is defined based on a fiber score, sweet spot score and DBS network score, which makes the analysis slightly more complex. We have published a multitude of studies that the present one builds upon (e.g., Horn 2017 *Annals of Neurology*, Al-Fatly 2019 *Brain*, Li 2020 *Nature Communications*, Li 2019 *Biological Psychiatry*) each building on the same concepts (alongside methodological work, e.g., Horn 2014, 15, 17 & 19, Ewert 2018 & 19, Treu 2020 and Wang 2021, all published in *NeuroImage*). We hope that the added sections (see below) may further explain our concepts to readers unfamiliar with DBS / this body of the literature.*

We have now added simple step-by-step instructions that explain all concepts to the supplementary material (which are also reproduced below in the response to the reviewer).

After reading the paper (and supplemental files) three times, I cannot tell what the input data was to the predictive model. It is unclear if individual resting state data from each subject was used or if the functional data was used from some other study. Functional data from the AD patient groups is not described, and supplemental files state only that 1.5T structural MRI data was acquired. There is reference to 1,000 healthy subjects functional data, “normative connectome” and “connectivity fingerprints” but it is never described what these are and how these data were generated. There is also a description of a DWI dataset acquired from a single subject, but no description of the DWI data from each AD patient. It seems that the only MRI data acquired from each subject was the T1w/T2w volumes and no DWI or fMRI data. This is very misleading as the narrative makes it seem as if there is DWI and fMRI from each subject that went into the predictive model. If there is individual level DWI and fMRI data, these data need to be described (acquisition parameters, preprocessing methods, quality checking).

*We are very sorry this was misleading and further emphasized the concept of using normative connectomes. Indeed, no connectivity data was acquired in these patients (and cannot be easily done due to the indwelling hardware). As in numerous prior studies (e.g., see Horn & Fox 2020 *NeuroImage* for a review), to investigate DBS effects on networks, we have registered DBS electrode data with normative connectomes acquired in i) an ultra-high-resolution scan of a single brain scanned for 18 hours to acquire a precise structural connectome (at 760 um isotropic resolution) defining the tracts in the models and ii) a cohort of 1,000 subjects from the Genomic Superstruct Project for functional connectivity.*

By doing so, we are asking the following question:

The modulation of which network in the average human brain would be associated with optimal improvements in Fx-DBS for AD.

This addition, “in the average human brain”, is crucial.

An independent research question could be, whether the same networks are associated in individual patients. Due to the lack of dMRI or rsfMRI scans in this rare patient population, we cannot ask this question. However, we have investigated this question in other diseases (Parkinson’s and OCD) where individual connectivity data was available (e.g., Wang et al. 2020, NeuroImage; Baldermann et al. 2019, Biological Psychiatry). We could show that results from normative connectomes and patient-specific or disease-matched connectomes were comparable in these diseases. In the present study, we showed that the conclusions drawn from one cohort (training) can estimate variance in outcomes of a second cohort (test), i.e., that the conclusions seem meaningful to make estimates for unseen data in the case of structural connectivity.

We have further emphasized the use of normative connectomes in the abstract and methods, as well as discussion:

“Using normative structural and functional connectivity data, we demonstrate that stimulation of the circuit of Papez and stria terminalis robustly associated with cognitive improvement ($R = 0.45$, $p = 0.031$). On a local level, the optimal stimulation site resided at the direct interface between these structures ($R = 0.33$, $p = 0.016$). Finally, modulating specific distributed brain networks accounted for optimal outcomes ($R = 0.38$, $p = 0.006$). Findings were robust to multiple cross-validation designs and may now define an optimal network target which could potentially guide refinement of DBS surgery and programming.” – Abstract, p. 3

“Model definition (Figure 1A): Whole brain structural connectivity profiles seeding from bilateral E-fields were calculated using a state-of-the-art multi-shell diffusion-weighted imaging dataset acquired across 18 scanning hours of a single individual at 760 μm isotropic resolution⁴¹ using the generalized q-sampling approach (default parameters) and whole-brain tracking (default parameters) as implemented in DSI studio⁸⁴. The patients were distributed into two cohorts: Training ($N = 28$) and Test ($N = 18$). For each subject of the training cohort, fibers traversing each voxel of the E-field were selected from the 5 million tracts in the normative connectome and projected to a voxelized volume in MNI space. Each of these fibers were weighted according to the E-field magnitude at each voxel, considering only fibers that traversed > 20% of stimulation volumes with an E-field magnitude > 0.36 V/mm. Each fiber was then appointed an R-value dependent on the Spearman correlation between its weighting and the respective clinical outcome scores across the group, i.e., a high R-value indicates that the modulation of that tract is associated with clinical improvement. Given the mass-univariate nature of this approach (and subsequent alpha-error accumulations), the resulting correlation coefficients were not considered significant, but were rather used to discriminate and visualize a specific set of bundles that was later validated by estimating clinical outcome in out of sample data (Figure 1, table S5).” – Methods, p. 25-26

We have further emphasized this concept in our limitations section:

“Moreover, we must emphasize that conclusions about connectivity profiles associated with optimal outcomes were based on normative connectivity data acquired in healthy participants. While this concept has led to meaningful and robust models in other cohorts^{12–14,38,71}, conclusions about networks prevalent in the individual DBS patients may not be drawn. However, models describing optimal connectivity based on normative

vs. disease-matched vs. patient-specific data were comparable in other diseases, such as Parkinson's Disease and OCD^{38,70}.” – Discussion, p. 21-22

“We used a normative whole-brain connectome calculated from an unprecedentedly high-resolution in-vivo dMRI dataset that was acquired across a total scan time of 18 hours at 760 μm isotropic resolution on specialized MR hardware⁷⁰, as for network mapping, a connectome obtained from rs-fMRI data from 1000 healthy subjects was used to inform regions co-activated with the stimulation volumes of each patient, allowing an identification of circuits that could be involved in clinical changes when modulating the fornix. We must emphasize that conclusions about connectivity profiles associated with optimal outcomes were based on normative connectivity data acquired in healthy participants. While this concept has led to meaningful and robust models in other cohorts^{12–14,38,71}, conclusions about networks prevalent in the individual DBS patients may not be drawn.” – Discussion, p. 22

We have further added a larger supplementary section detailing the normative connectomes (scan parameters, references, and data sources):

“Normative Connectomes: Underlying Data” – Supplementary Material

Table S4. Specification of normative connectome data. Abbreviations: TR = Repetition time, TE = Echo time, FOV = Field of view, BOLD = Blood oxygenation level-dependent, EPI = Gradient-echo echo-planar imaging, FA = Flip angle

Connectome	Scan parameters	References	Data sources
Structural: In vivo human whole-brain Connectom diffusion MRI dataset at 760 μm isotropic resolution	Scanner: MGH-USC 3T Connectom. Maximum gradient strength of 300mT/m and maximum slew rate of 200 T/m/s, custom-built 64-channel phased-array coil. gSlider-SMS sequence. gSlider encoding: 5 MB factor: 2 R_{inplane} factor: 3 Acquisition: Axial (PE along AP/PA) TR/TE: 3500/75 ms FOV: 220.0 \times 218.5 mm Acquisition matrix: 290 \times 288 Acquired slices: 190 Slice thickness: 0.76 mm Effective echo spacing: 0.34 ms Readout bandwidth: 1150 Hz/Pixel Phase partial Fourier: 6/8 b-values: 1000, 2500 s/mm² 144 (b0), 420 (b1000), 840 (b2500) w/AP/PA (total 2808 volumes) Total acquisition time: ~14.5 hours	Wang et al. (2021) Sci. Data¹	9 two-hour scan sessions 1 healthy subject
Functional: The organization of the human cerebral cortex estimated by intrinsic functional connectivity	Scanner: 3T Tim Trio scanners (Siemens, Erlangen, Germany) 12-channel receive coil array, Gradient -echo echo-planar imaging (EPI) sequence sensitive to BOLD contrast. Acquisition: Slices aligned to anterior commissure-posterior commissure plane EPI parameters TR/TE: 3000 ms/30 ms FA: 85°, 3 \times 3 \times 3-mm voxels FOV: 216 47 axial slices collected with intervalued acquisition, no gap between slices 6.2 minute-functional run (124 timepoints)	Yeo et al. (2011) J. Neurophysiol² Holmes & Buckner	Resting-state fMRI data from 1,000 health subjects (average 1.7 runs per subject)

The authors make claims regarding generalizability and robustness, however these claims are not supported by the statistical framework presented to the reader. Given the strong clinical claims (“Potentially, our results can be useful to guide DBS programming in existing patients with fx-DBS and potentially inform surgical targeting in AD within future investigational trials”), this lack of detail is very concerning and needs to be addressed before consideration of publication in any journal.

We have deleted the clinical claim and have now adjusted the analysis concept based on the community standards developed by Poldrack et al. (Poldrack et al. 2017 JAMA Psychiatry)

Moreover, as mentioned, code & data is now made openly available:

“Data Availability

Data used for the described analyses is openly available on OSF (<https://osf.io/bckuf>). The resulting tract atlas, sweet spot and fMRI network pattern are openly available within Lead-DBS software (www.lead-dbs.org).

Code availability

All code used to analyze the dataset is openly available within Lead-DBS/Connectome software (<https://github.com/leaddbs/leaddbs>). Code to reproduce figures is openly available on OSF (<https://osf.io/bckuf>).” – Methods, p. 28-29

Without much to go off of in terms of evaluating the machine learning framework, my main concerns are regarding the lack of model comparison across all of the test sets and different cross validation strategies and the lack of clarity about what data each subject contributed to the predictive model.

The following supplementary section was added to narratively clarify the three concepts further (also see figure 1 and methods sections DBS fiber filtering, Optimal Stimulation Sites (Sweetspot Analysis) and DBS Network Mapping):

“In all three models, each patient contributed their relative improvement of ADAS-cog-11 scores (before surgery, one year after surgery).

Beyond that, each model (i) tracts, ii) sweetspots and iii) functional networks) was run independently from one another.

- *i) For tracts, each patient contributed the peak E-field amplitude that each tract of the normative connectome was modulated by.*
- *ii) For sweetspots, each patient contributed the modeled electric field in MNI space (represented as a NIFTI volume).*
- *iii) For functional networks, each patient contributed a (normative) rs-fMRI map seeding from the individual patient (“connectivity fingerprints”).*

Then, the three models created a i) combination of tracts ii) optimal target (sweetspot), and iii) functional network profile associated with optimal clinical improvements.

- *i) For tracts, this was achieved by rank correlating the modulation amplitude imposed on each tract with clinical improvements across the set of patients. This led to an R-value for each tract, denoting how well its modulation correlated with clinical improvements (the concept was introduced in Irmen et al. 2019 Annals of Neurology).*
- *ii) For sweetspots, this was achieved by rank correlating each voxel with clinical outcomes across the set of patients. This led to an R-map denoting how well modulations of specific voxels correlated with clinical outcomes (the concept was introduced in Horn et al. 2022 PNAS).*
- *iii) For functional networks, this was achieved by correlating the voxel values of connectivity fingerprints with clinical improvements across the set of patients. This led to an R-map denoting how well connectivity estimates between stimulation sites and each voxel in the brain correlated with clinical outcomes (the concept was introduced in Horn et al. 2017 Annals of Neurology).*

Finally, data was cross-validated within the three models:

- i) For tracts, this was achieved by rank correlating the impacts of the E-Fields of a left-out patient on all tracts and their R-values. This led to a fiberscore denoting how specifically the E-Field in a left-out patient modulated tracts associated with optimal outcomes (the concept was introduced in Horn et al. 2022 PNAS).
- ii) For sweetspots, this was achieved by spatially correlating the E-Fields of an unseen patient with the R-map model. This led to a sweetspot score denoting correlation coefficients of agreement between the actual stimulation field and an “optimal” stimulation field (represented by the R-map; the concept was introduced in Horn et al. 2022 PNAS).
- iii) For functional networks, this was achieved by spatially correlating the functional connectivity fingerprints with the R-map model. This led to a network score denoting correlation coefficients of agreement between the actual network profile and an optimal network profile (represented by the R-map; the concept was introduced in Horn et al. 2017 Annals of Neurology).” – supplementary material, p. 18-19

Several statements made throughout the paper are concerning to me and are suggestive that proper statistical inference has not been performed and therefore the analysis/results do not support the conclusions of the paper.

“Purely visual test-retest comparison of results”

We agree and changed the wording:

“As further evaluation, we calculated the predictive tract model based on the training-, test- and combined cohorts, separately. This allowed a direct comparison of results calculated in each cohort by visual inspection, and overlaid the identified bundle with structures of interest from atlases in MNI space^{39,47} (Figure S4).” – Results, p. 9

No permutation testing or any null hypothesis testing. Only reporting cross validated Pearson correlation and p-values that are averaged across folds. There are no error bars/confidence intervals, or accuracy evaluation metrics (i.e., mean squared error, median absolute error) reported. This is a huge red flag

<https://www.sciencedirect.com/science/article/pii/S1053811917305311> & <https://jamanetwork.com/journals/jamapsychiatry/article-abstract/2756204>)

We now report numerous additional metrics, such as median absolute deviation (MAD) and root-mean square deviation (RMS), as well as coefficient of determination R^2 , results from permutation testing and include code & data to the submission for maximal transparency. All code and data to reproduce figures is now included with the submission and will be openly made available on OSF. The following paragraphs were added:

“For each of the models, the stimulation volume of each patient was considered the core of the analysis; for fiber filtering, streamlines from a normative structural connectome that traversed the volumes were considered for further steps; for sweetspot analysis, areas of interest were determined based on voxels occupied by stimulation volumes of the patients; finally, for network mapping, functionally connected areas to the stimulation volume of each patient were obtained from a functional normative connectome. Details for each method are specified in the following sections.

DBS fiber filtering

Model definition (Figure 1A): Whole brain structural connectivity profiles seeding from bilateral E-fields were calculated using a state-of-the-art multi-shell diffusion-weighted

imaging dataset acquired across 18 scanning hours of a single individual at 760 μm isotropic resolution⁴¹ using the generalized q-sampling approach (default parameters) and whole-brain tracking (default parameters) as implemented in DSI studio⁸⁴. The patients were distributed into two cohorts: Training (N = 28) and Test (N = 18). For each subject of the training cohort, fibers traversing each voxel of the E-field were selected from the 5 million tracts in the normative connectome and projected to a voxelized volume in MNI space. Each of these fibers were weighted according to the E-field magnitude at each voxel, considering only fibers that traversed > 20% of stimulation volumes with an E-field magnitude > 0.36 V/mm. Each fiber was then appointed an R-value dependent on the Spearman correlation between its weighting and the respective clinical outcome scores across the group, i.e., a high R-value indicates that the modulation of that tract is associated with clinical improvement. Given the mass-univariate nature of this approach (and subsequent alpha-error accumulations), the resulting correlation coefficients were not considered significant, but were rather used to discriminate and visualize a specific set of bundles that was later validated by estimating clinical outcome in out of sample data (Figure 1, table S5).” – Methods, p. 25-26

“Next, we used the fiber model calculated on the complete training cohort (N = 28) to estimate clinical outcomes in patients from the test cohort (N = 18), which had been left as a completely naïve hold-out set (Figure S4B). This cross-cohort-prediction revealed a significant relationship (R = 0.45 at p = 0.031, R² = 0.102, RMS = 41.621, MAD = 25.452; Figure 2C) indicating robustness of the generated model. It should be noted that for out-of-sample testing, the coefficient of determination R² is computed based on the sum of squared errors, and not by squaring the correlation coefficient⁴⁶.” -Results, p.8

“As a final validation step, we carried out a leave-one-out cross validation across the whole cohort which led to an R = 0.66 at p < 10⁻¹⁶, RMS = 50.32, MAD = 33.23 between estimated fiber scores and empirical improvements. Further cross-validation k-fold designs led to similar results (3-fold: R = 0.44 at p = 0.002; 5-fold: 0.50 at p < 10⁻¹⁶; 7-fold: R = 0.48 at p = 0.001; and 10-fold: R = 0.52 at p < 10⁻¹⁶, see Figures 5 and S7 for additional metrics).” -Results, p. 9

“Instead, spatial maps consisting of sweet- and sour-spots were cross-validated across the entire cohort in a leave-one-patient-out design, which led to significant results (R = 0.33 at p = 0.016, RMS = 50.60, MAD = 27.94). Further cross-validation designs led to similar results (3-fold: R = 0.27 at p = 0.037; 5-fold: R = 0.30 at p = 0.016; 7-fold: R = 0.39 at p = 0.005; 10-fold: R = 0.33 at p = 0.011, Figures 5 and S7).” -Results, p.11

“To validate these results, we again carried out leave-one-out (R = 0.38 at p = 0.006, RMS = 48.69, MAD = 30.99) and several k-fold cross-validation designs (3-fold: R = 0.32 at p = 0.015; 5-fold: R = 0.14 at p = 0.147; 7-fold: R = 0.44 at p < 10⁻¹⁶; 10-fold: R = 0.29 at p = 0.020, Figures 5 and S7).” -Results, p. 13

R², RMS and MAD were added to figure 2, and an additional figure showing an in-fold analysis of cross-validation results of fiber filtering (figure S3).

A

FIBERTRACT OBTAINED FROM TRAINING COHORT - 28 PATIENTS

B LEAVE ONE OUT CROSS VALIDATION - TRAINING - 28 PATIENTS

C VALIDATION IN OUT OF SAMPLE DATA - TEST - 18 PATIENTS

Figure 2. Validation of tract models predictive of clinical improvements as evaluated using ADAS-Cog 11. A) Left: Optimal set of tracts to be modulated as calculated from the entire training cohort ($N = 28$ subjects). Right: permutation analysis calculated on the entire training cohort. B) Top left: stimulation volume of a patient with top clinical improvement overlapping the tracts associated with optimal clinical improvements (calculated leaving out the subject, $N = 28 - 1 = 27$ subjects). Fibers displayed in white correspond to the portion of optimal fibers intersecting with the patient's stimulation volume. Bottom left: Same analysis carried out with a poor-responding example patient. Right: Cross-validation within the training cohort using a leave-one-out design (top, $R = 0.69$ at $p < 10^{-16}$) and within-fold analysis (bottom). The two example patients are marked in the correlation plot with circles. C) Optimal tracts calculated from the entire training cohort (as shown in panel A, $N = 28$) were used to cross-predict outcomes in $N = 18$ left out patients of the test cohort ($R = 0.45$, $p = 0.031$). Left: two example cases from the test cohort are shown, a top responding patient's stimulation volume with corresponding connected (white) optimal fibers (defined by the training cohort); and a poor-responding patient's stimulation volume with corresponding connected (white) fibers. The two example patients are marked in the correlation plot with circles. Fiber tracts and example stimulation volumes were superimposed on slices of a $100\text{-}\mu\text{m}$, 7T brain scan in MNI 152space⁴⁸. Gray shaded areas represent 95% confidence intervals.

Figure S3. In-fold analysis from fiber filtering analysis on Training cohort showing absolute predicted error, root mean square deviation (RMS) and median absolute deviation (MAD) for each of the validation approaches.

RMS and MAD are now reported throughout Leave-one-out and k-fold analyses (Figures 5 and S7).

Figure 5. Results summary including the models from DBS fiber filtering, sweetspot mapping and network-mapping. The three levels of analysis were able to explain a similar amount of variance of clinical outcomes when analyzed in a circular nature (see scatterplots; ~16-19%) and led to significant cross-predictions of clinical outcomes across leave-one-patient-out and multiple k -fold designs. Three level analysis results were superimposed on slices of a brain cytoarchitecture atlas in MNI 152 space⁵⁰. Gray shaded areas represent 95% confidence intervals.

A FIBER FILTERING

Continuation Figure S7

Continuation Figure S7

Figure S7. In-fold analysis from summary showing absolute predicted error, root mean square (RMS) and median absolute deviation (MAD) for each of the validation approaches followed on fiber filtering (A), sweetspot (B) and network mapping (C) methods.

Reviewers' comments:

Reviewer #1 (Remarks to the Author):

The authors have done an admirable job addressing my comments. The manuscript and the dataset are valuable and should be considered for publication.

Reviewer #2 (Remarks to the Author):

The revised version has addressed all my comments and I believe it is ready for publication now.

Reviewer #4 (Remarks to the Author):

The resubmission has sufficiently addressed Reviewer 3 comments. The following two minor points should be addressed further:

- Small sample size for machine learning experiments should be added to limitations
- As authors have stated in their response this study should be tested as statistical analysis and not as machine learning experiments. —Therefore, use of “test” and “unseen” data should be removed, as it is slightly misleading given the sample size.

Reviewer #1 (Remarks to the Author):

The authors have done an admirable job addressing my comments. The manuscript and the dataset are valuable and should be considered for publication.

We would like to thank the reviewer for their positive evaluation.

Reviewer #2 (Remarks to the Author):

The revised version has addressed all my comments and I believe it is ready for publication now.

We would like to thank the reviewer for their positive evaluation.

Reviewer #4 (Remarks to the Author):

The resubmission has sufficiently addressed Reviewer 3 comments. The following two minor points should be addressed further:

- Small sample size for machine learning experiments should be added to limitations
- As authors have stated in their response this study should be tested as statistical analysis and not as machine learning experiments. —Therefore, use of “test” and “unseen” data should be removed, as it is slightly misleading given the sample size.

We would like to thank the reviewer for their overall positive evaluation.

The following sentence was added to the Limitations section:

“Nevertheless, this sample size is considered small for machine learning approaches, thus, rigorous evaluation was performed to the results presented in this work, including cross-validation at several levels.” (Discussion, page 27)

Uses of terms “test” and “unseen” were removed throughout the manuscript.